# Sensitivity analysis of factors influencing the ecology of mosquitoes involved in the transmission of Rift Valley fever virus

Jessica Furber[1]*, Sophie North[1], Martha Betson[1], Christophe Boëte [2]*, Daniel Horton[1,3], Giovanni Lo Iacono[1,4,5]

1 School of Veterinary Medicine, University of Surrey, Guildford, Surrey, United Kingdom, 2 ISEM, Univ Montpellier, CNRS, IRD, Montpellier, France, 3 Department of Virology, Animal and Plant Health Agency, Woodham Lane, Addlestone, United Kingdom, 4 Institute for Sustainability, University of Surrey, Guildford, Surrey, United Kingdom, 5 The Surrey Institute for People-Centred Artificial Intelligence, University of Surrey, Guildford, Surrey, United Kingdom

* j.r.furber@surrey.ac.uk (JF); christophe.boete@ird.fr (CB)

## Abstract

Vector-borne diseases are a major global health concern, with Rift Valley fever (RVF) serving as a key example due to its impact on both human and animal health. Predicting and controlling such diseases requires understanding how environmental factors shape mosquito ecology. Due to mosquito abundance, distribution, and behavior being influenced by ecological conditions, identifying the drivers of these dynamics is essential for anticipating transmission risk. This study aims to assess the sensitivity of ecological factors governing the ecology of mosquitoes, using a deterministic, compartmental model of RVF transmission in Kenya. We conducted a local and global sensitivity analysis on ten model parameters: four species-specific parameters for each of the two mosquito species of interest (*Culex* spp. and *Aedes* spp.) and two parameters shared across species. The focal species-specific parameters were the area scanned before oviposition, the proportion of area suitable for egg laying, the number of eggs laid, and the maximum density of eggs. The two shared parameters represented livestock population size and the influence of livestock on vector fecundity and gonotrophic cycles. Parameter ranges and distributions were defined using a scoping literature review. Sobol sensitivity analysis was performed under two environmental scenarios: (i) constant temperature and water body area, and (ii) periodically varying temperature and water body area, implemented using a time-dependent Sobol framework. Results revealed species-specific differences in parameter influence. For *Culex* spp., uncertainty in the area scanned for oviposition was highly influential, while for *Aedes* spp., the pairing of the proportion of area that eggs are laid on and the maximum density of eggs emerged as dominant. These findings highlights the need for improved empirical data on spatial oviposition patterns across water bodies, as current evidence remains limited. By identifying the ecological parameters that most critically shape mosquito population dynamics and/or influence

**Data availability statement:** The code used for the analysis is available at the Github repository: https://github.com/JFurber/Sensitivity-Analysis-RVFV The code includes all scripts necessary to reproduce the analysis and results presented in the paper. The analysis was performed using R, with specific packages or libraries, which are documented in the repository.

**Funding:** This work was co-funded by the European Union's Horizon Europe (Project 101136346 to GLI; Project 101136346 to JF) and generously funded by the Longhurst Legacy (SN only). The funders had no role in study design, data collection and analysis, decision to publish, or preparation of the manuscript. Views and opinions expressed are those of the author(s) only and do not necessarily reflect those of the European Union or the European Research Executive Agency. Neither the European Union nor the granting authority can be held responsible for them.

**Competing interests:** The authors have declared that no competing interests exist.

model outputs, and thereby the transmission potential of RVF, this work supports more targeted vector surveillance and strengthens public health decision-making in RVF-endemic regions.

## Author summary

Rift Valley fever is a disease spread by mosquitoes that affects both people and animals. Understanding what drives mosquito populations can help us predict and control the spread of Rift Valley fever. In this study, we used a published mathematical model to conduct a sensitivity analysis to explore how different parameters within the model affect mosquito numbers. In total, we investigated ten parameters, where the analysis emphasizes the role of water bodies used by mosquitoes to lay their eggs. We found that this factor plays a major role in how many mosquitoes are present. However, there is still a lot we do not know about exactly where mosquitoes choose to lay their eggs. More research in this area could improve how we predict and manage Rift Valley fever and other mosquito-borne diseases outbreaks. Our findings highlight which environmental details matter most, and they can help guide better strategies for mosquito control and disease prevention.

## Introduction

Rift Valley fever (RVF) is a mosquito-borne viral disease caused by the Rift Valley fever virus (RVFV), which is currently considered a priority for research and development in emergency contexts by the World Health Organization [1]. First identified in Kenya's Rift Valley in 1931 [2,3], RVFV has since spread across Africa, with outbreaks in Saudi Arabia and Yemen in 2000 raising concerns about its geographic expansion [4]. Transmission is strongly seasonal, driven by environmental conditions that affect mosquito populations, making prediction both possible and essential. However, existing models struggle to determine which parameters most influence vector dynamics due to limited data availability and model complexity.

Kenya has experienced significant RVF activity, including a severe 2007 outbreak that devastated pastoral communities in the northeast. In the Garissa District alone, an estimated 300,000 animals died, triggering economic hardship across the livestock sector [5]. While RVFV primarily affects domestic ruminants, causing abortions and high mortality in young animals, it also infects humans, occasionally leading to serious complications such as ocular damage, renal failure, or hemorrhagic fever [6–8]. These impacts highlight the urgent need for effective predictive models to support RVF control.

Given the impact of RVFV on both humans and livestock, understanding its mosquito vectors is crucial for disease modelling and control efforts. Pepin *et al.* (2010) [9] described two distinct groups of RVFV vectors, referred to as 'reservoir/

maintenance vectors' (i.e., primary vectors) and 'epidemic/amplifying vectors' (i.e., secondary vectors). The former group is thought to primarily consist of floodwater *Aedes* spp. mosquitoes, while the latter group is believed to comprise mainly *Culex* spp. mosquitoes, though *Mansonia* and *Anopheles* may be involved as well [10].

One key factor influencing RVFV transmission is the breeding behavior of its primary mosquito vectors. In some regions of Kenya, floodwater *Aedes* spp. mosquitoes typically lay their eggs in dambos: grass-covered, shallow depressions which become waterlogged after rainfall [11]. Crucially, *Aedes* spp. eggs must be submerged in water to hatch. During periods of particularly low rainfall in these areas, *Aedes* spp. eggs will build up along the perimeter of the dambos, where they can survive desiccation for many months in a state called diapause [12]. Numbers of adult *Aedes* spp. mosquitoes may diminish as the eggs lie dormant. When rain does eventually occur, considerable quantities of adult mosquitoes can emerge in one go as many submerged eggs are allowed to develop simultaneously. A proportion of these may be infected with RVFV, due to the ability of *Aedes* spp. mosquitoes to vertically transmit the virus to their eggs [13]. Thus, this mass emergence of adults sets up a scenario where RVFV outbreaks are particularly likely [14].

A deterministic, compartmental ecoepidemiological model of RVFV transmission [15] captures key disease-environment interactions, but lacks clarity on the relative influence of specific ecological parameters, prompting this study's focus on sensitivity analysis to better understand the drivers of mosquito abundance. Our analysis is conducted in a simplified setting, focusing on vector ecology and initially excluding fluctuations in temperature and water body area. We focus on four key parameters for each species with uncertain values in the literature, along with two additional parameters that captures the influence of livestock on vector dynamics, aiming to determine which have the greatest influence on model outcomes. Specifically, we investigated: typical area scanned by mosquito flyers in relation to the known extent of water-bodies, proportion of water body area/soil in which eggs are laid, number of eggs per batch, maximum surface density of eggs, and also included the number of livestock used and a parameter from the mathematical model that quantifies the impact of the livestock on vector fecundity and gonotrophic cycles. The sensitivity analysis was conducted only on independent parameters whose values are not derived from other quantities. In contrast, parameters such as the vector-to-host ratio and developmental rates, although potentially sensitive, are defined by specific mechanistic relationships with temperature, the availability of water bodies, and/or host abundance. These relationships constrain their variation, meaning that changes in these parameters are predictable and not random or arbitrary. The focus of this work is therefore to identify the most influential parameters and those with high uncertainty, which warrant further attention. Parameter distribution information can also inform priors for future Bayesian inference and help prioritize influential parameters when high dimensionality limits inference.

Our findings highlight a significant knowledge gap surrounding spatial patterns of mosquito oviposition across larger bodies of water. If more accurate estimates of these parameters were available, this could improve both the accuracy and usefulness of mathematical models of mosquito-borne disease such as ours.

## Materials and methods

### Literature review search methodology

A scoping review was carried out to obtain (for each of the parameters of interest) a range of potential values and a distribution within that range. A literature search was conducted on PubMed to identify studies related to the reproductive behavior and ecology of key RVFV vector mosquitoes in Kenya. The PubMed query used was: (oviposit* OR fecundity) AND ((*Aedes* AND (*circumluteolus OR mcintoshi OR ochraceus OR pembaensis OR sudanensis OR dentatus*)) OR (*Culex* AND (*poicilipes OR quinquefasciatus OR univittatus OR zombaensis OR bitaeniorhynchus OR pipiens*))) AND (behaviour OR behavior OR ecology). These terms were chosen to capture oviposition- and fecundity-related studies across important *Aedes* and *Culex* spp. listed in the search, which are recognised as major RVFV vectors in Kenya [16]. We searched literature from 1966 to 2024 and considered only articles written in English.

**PLOS** Neglected Tropical Diseases

The initial search yielded 169 results. Screening of titles reduced this number to 68; at this stage, studies referring to oviposition deterrence or attraction involving chemicals, pheromones, hormones, pond dyes, plant-derived essential oils, or the influence of aquatic vegetation on oviposition behaviour were excluded. Similarly, papers focusing on the anatomy or physiology of mosquito sensory responses, or on comparisons of egg collection methods, were removed. Further examination of abstracts narrowed the selection to 48 relevant studies, and after a detailed review of the full texts, eleven papers were ultimately included in the analysis. These studies provided critical insights into the reproductive and ecological traits of RVFV vector species, aiding in a more profound understanding of their role in disease transmission dynamics. Eight more publications were included from citation searches. Regarding the impact of livestock on mosquito oviposition behaviour, to the best of our knowledge, the only direct research available is one study in the context of Chagas disease for triatomines [17]. We referenced this study due to the lack of more directly relevant literature. While there are other studies examining the influence of livestock on mosquito ecology, most focus on the attraction of mosquitoes to livestock through chemical cues, such as $CO_2$ or other volatiles. However, this was beyond the scope of our current study.

The results of this review have been used to carry out a sensitivity analysis using Sobol's method of sensitivity indices [18]. We then further investigate the mechanisms behind these indices (see below).

## Mathematical model

The model used in this analysis is based on the framework developed in a previous study [15], which was largely based on the stage-structured, population dynamics model of Otero *et. al* [19]. The *Culex* spp. mosquito population is divided into six stages: eggs, larvae, pupae, nulliparous females (adult females that have not yet laid eggs), flyers (females searching for oviposition sites), and parous females (those that have laid eggs). We assume that *Culex* spp. mosquitoes lay their eggs in an inner zone near the edge of the water body. The model does not explicitly account for the spatial geometry of water bodies; instead, this inner area is represented as a proportion of the total, in general time-varying, water body area, denoted by $\kappa^{Culex}$. For clarification, in the case of a perfect circular water body, this inner area can be represented as an annulus, the ring-shaped area between two concentric circles, corresponding to the zone around the water body edge where egg laying occurs. Let the radial thickness (i.e., the difference between the outer and inner radii) be denoted by $\Delta r$. The area of the annulus is then:

$$\pi(r + \Delta r)^2 - \pi r^2 = \pi(\Delta r^2 + 2r\Delta r). \qquad (1)$$

This quantity is generally proportional to a combination of both the total area and the perimeter of the water body. If $\Delta r$ were constant regardless of the water body size, then the annulus area would increase linearly with the perimeter. However, we think it is more realistic to assume that $\Delta r = \alpha r$, meaning that the annulus thickness scales with the inner radius (i.e., compared to small pool, larger water bodies have a wider oviposition zone). Substituting this relationship gives an annulus area of

$$\pi(\alpha^2 + 2\alpha)r^2, \qquad (2)$$

where $(\alpha^2 + 2\alpha)$ corresponds to $\kappa^{Culex}$. The *Aedes* spp. life cycle is similar but distinguishes between immature and mature eggs, as *Aedes* spp. females lay eggs in moist soil above the waterline rather than directly on the water surface. This zone is represented as a proportion of the total water body area, denoted by $\kappa^{Aedes}$ (assuming that larger water bodies have a wider zone of moist soil around the edge), with hatching occurring once the eggs are submerged after a minimum desiccation period. In reality, $\Delta r$ is likely a complex function of multiple factors, including the area and shape (and therefore the perimeter) of the water body, as well as environmental variables such as temperature and evaporation rate. The current model can be adapted for situations in which the role of the perimeter becomes dominant by redefining the terms

$\rho_A$ and $\rho_C$ as the maximum *linear* density of eggs rather than surface density, *i.e.*, assuming that mosquitoes lay their eggs along a narrow band that can be approximated as a one-dimensional line rather than a surface. At present, however, no estimates of these linear densities are available.

Population dynamics across stages are represented by a system of differential equations describing temporal changes in the abundance of the different stages of mosquitos. Key factors include oviposition, stage-specific mortality, and development rates governing transitions between stages. Temperature affects mortality and development rates [19,20]; the availability and amount of water bodies, as proxies for breeding sites, influences oviposition and the transition of flyers to adults upon locating suitable sites [15]; livestock density, as a source of blood meals, affects egg batch size and the gonotrophic cycle (biting rate) [17].

The full mathematical model is presented in the Supplementary Information of [15], where *Culex* and *Aedes* spp. population dynamics are detailed in the first and second equations, respectively. The influence of temperature, water bodies, and livestock density on ecological parameters is specified in Equations. (3)–(23). The parameters of interest for this study are detailed next.

## Parameter quantification

The model is set up to receive daily empirical data for air temperature and water body area. Firstly, to allow for more transparent analysis, we removed some layers of complexity, imposing a constant temperature and constant water body area. However, in the event that this had an effect on the results, we set up three levels for temperature and water body levels. A further parameter representing the probability of mosquitoes to find and feed on the particular host species of interest (i.e., detection probability, denoted as $p_f$ in [15] and *prop_find* in the code) was also assigned multiple levels. For each of these three additional parameters, 'low', 'medium' and 'high' values were selected, with the temperature and water body values based on empirical datasets previously used for Kenya (see Supplementary Information [15]). The values for the parameter $p_f$ were arbitrarily selected. These values are detailed in Table 1.

Secondly, since the oviposition rate of *Aedes* spp. mosquitoes also relies on the fluctuations of the water body areas, i.e., the eggs are laid on soil and the flooding allows them to hatch, then the analysis was completed using periodic functions to denote the temperature and water bodies, and using a time-dependent Sobol analysis to account for the changes in parameter influence over time as environmental conditions fluctuate. This approach enables the sensitivity analysis to reflect how the impact of each parameter evolves in response to seasonal variation, particularly in relation to the timing and extent of water body inundation that governs egg hatching dynamics. These periodic functions approximated, based on empirical data, respectively, in [15] as

$$T(t) = T_m + T_A \cos\left(\omega_T t + \phi_T\right), \tag{3}$$

$$S^P(t) = S_m^P + S_A^P \cos\left(\omega_s t + \phi_S\right), \tag{4}$$

**Table 1. Choice of constant low, medium and high parameter values for sensitivity analysis.**

| Parameter | low | medium | high |
|---|---|---|---|
| Temperature (°C) | 21 | 25 | 29 |
| Water Body Area (m²) | 10 000 | 17 500 | 25 000 |
| Detection Probability, $p_f$ | 0.01 | 0.1 | 0.5 |

where $\omega_T$ and $\omega_S$ are the frequencies of oscillations in temperature and surface areas of water bodies; the terms $T_m$ and $S_m^P$ represent the mean temperature and mean surface area of water bodies during periods $2\pi/\omega_T$ and $2\pi/\omega_S$, respectively; $T_A$ and $S_A^P$ are the maximum amplitudes in the oscillations; and $\phi_T$ and $\phi_S$ are the respective phases.

As mentioned in the introduction, the following parameters were reviewed in the sensitivity analysis: i) the typical area scanned by mosquitoes before finding an oviposition site, ideally measured from the movement of gravid females between blood-feeding and oviposition sites, but often inferred from general mosquito dispersal distances due to limited information ($\mathcal{A}_C$ and $\mathcal{A}_A$ for *Culex* and *Aedes* spp. mosquitoes, respectively); ii) the proportion of the water body area that adult female *Culex* spp. mosquitoes lay eggs on ($\kappa^{Culex}$) and the proportion of soil that adult female *Aedes* spp. mosquitoes lay eggs on ($\kappa^{Aedes}$, denoted as $\kappa^{Aedes}$ in [15]); iii) the number of eggs laid per batch by adult female *Culex* and *Aedes* spp. mosquitoes ($b_C$ and $b_A$, respectively); iv) the maximum density of eggs in the water body for mosquitoes to lay eggs ($\rho_C$ and $\rho_A$, respectively, for *Culex* and *Aedes* spp.); v) livestock total; vi) the impact of the livestock on vector fecundity and gonotrophic cycles ($q$). Here and throughout, we refer to this parameter as 'livestock–mosquito reproduction coefficient'

The key ecological parameters of interest are nested within equations, such that the area scanned by flyers ($\mathcal{A}$) appears in equations governing oviposition rate ($\eta$, Supplementary Information of [15], Eq. (3)), the density of eggs per unit area ($\rho$) and the proportion of area that eggs are laid on ($\kappa$) appear in the equations governing the carrying capacity ($K$, Supplementary Information of [15], Eq. (5) and (9)), and the number of eggs per batch ($b$) appear in the Supplementary Information of [15], Eq. (6) and (10). Here and throughout, the absence of superscripts on parameter symbols indicates that the parameter applies to both *Culex* spp. and *Aedes* spp.

## Sobol sensitivity analysis

The Sobol method of sensitivity analysis quantifies the proportion of variance in a model output attributable to a specific parameter [21,22]. A lower sensitivity index indicates that the parameter contributes minimally to output variations, whereas a higher sensitivity index signifies a greater influence on model variability. When a parameter has a low sensitivity index, rough estimations may be sufficient and/or the associated process can be removed from the model to simplify the analysis, whereas high-sensitivity parameters require more accurate estimation.

In general, to conduct the Sobol sensitivity analysis, we first defined parameter ranges and their probability distributions using values reported in the literature. Quasi-random sampling methods were then used to generate parameter sets, providing uniform coverage of the parameter space and ensuring robust exploration of possible model behaviours [18]. The model was run repeatedly using these sampled parameters to capture how variation in inputs propagates to the outputs.

Firstly, simulations were performed under constant temperature and water-body conditions. Each run spanned ten years, with the first nine years treated as a stabilization period to allow the system to reach equilibrium. Sensitivity was evaluated using the adult mosquito population summed over the final 365 days, providing a single, stabilized output metric for the index calculations.

After this baseline assessment, we conducted a time-dependent analysis to investigate how parameter influence evolved throughout the simulation. Using the function *sobol_ode* from the *sensobol* package (version 1.1.5) [23], model outputs were evaluated at monthly intervals over two post-transient years, following the same nine-year stabilization phase. This approach allowed us to characterise the temporal stability of the sensitivity structure and identify periods in which certain parameters exerted greater or lesser influence.

Once simulations across all scenarios were completed, Sobol indices for first-order, second-order, and total-effect were computed using the *sensobol* function *sobol_indices*. The first-order indices represent the direct contribution of each parameter to the output variance, indicating how much influence a single parameter has when considered independently. In contrast, the total-effect indices account for both the direct effects and any interactions with other parameters, providing a more comprehensive view of how each parameter, including its interactions, influences the model's behavior. For example, if temperature has the highest first-order Sobol index, it suggests that it is a strong independent driver of the system.

In contrast, if rainfall exhibits a high total-order index but a low first-order index, this indicates that its influence arises mainly through interactions with other variables, such as vegetation. Meanwhile, if host availability has a very low Sobol index, it may not be a significant factor in the system, at least under the current assumptions. Finally, the second-order indices describes the variance in the output due to interactions between two input parameters (see S4 Appendix for full details of the method and results).

The reliability and convergence of the Sobol indices is heavily dependent on the number of sample runs. The higher the number of samples, the more likely it is that the indices will converge on the 'true' value. However, the more samples run, the higher the computational cost and time taken. A commonly referenced approach to choosing the number of sample runs suggests that the total number of model evaluations should be proportional to the number of input parameters ($m$) plus two, multiplied by the number of baseline samples per parameter ($N$), following the formula $N \times (D+2)$, where $N$ is equal to one thousand [18].

To assess the robustness of the Sobol sensitivity indices, a convergence analysis was performed by evaluating index estimates across increasing sample sizes for a *Culex* spp. scenario (S1 Fig). In general, both first-order and total-effect indices exhibited stabilization beyond a sample size of $N = 1000$, with minimal variation observed in the total-effect indices thereafter. This indicates that the sensitivity estimates are sufficiently stable and not unduly influenced by sampling noise at higher $N$. Some residual variation in the total-effect indices may arise due to their dependence on higher-order interactions and non-linear effects, which require more extensive sampling to resolve accurately. Unlike first-order indices, total-effect indices capture the full contribution of each parameter, including interactions with others, making them more sensitive to sampling density and model complexity. Based on this convergence behavior, it is justified to use a sample size of $N = 2000$ for the final analysis, ensuring reliable estimation of both first-order and total-effect indices. The full code used within this analysis on R [24], featuring the model and the Sobol analysis, is available via GitHub [25]. Next, we briefly describe the background of first-order and total-order indices.

**First order indices.** Let the parameter of interest be $Q_i$, for $i = 1, \ldots, N$, where $N$ is the number of parameters, $Y$ is the outcome of the model which depends on the parameters $Q_i$ (for instance, $Q_1$ is air temperature, $Q_2$ is water-bodies surface and $Y$ is the total abundance of mosquitoes over one season). The first-order Sobol sensitivity index $S_i$ quantifies the proportion of the total variance in the model output that is attributable solely to the variation in $Q_i$, while all other parameters are allowed to vary. In other words, it measures the reduction in model output variance if $Q_i$ were fixed, leaving the other parameters to vary freely. The first-order Sobol index for a parameter $Q_i$ is defined as the ratio of the variance of the conditional expectation $V(E[Y \mid Q_i])$ to the total output variance $V[Y]$, i.e.,

$$S_i = \frac{V[E[Y \mid Q_i]]}{V[Y]}.$$

(5)

Here, $E[Y \mid Q_i]$ represents the expected value of the model output when $Q_i$ is fixed and all other parameters are allowed to vary according to their distributions. Although the model used in this study is deterministic (i.e., it produces a single, fixed output for any given set of input parameters) Sobol sensitivity analysis treats the input parameters as random variables drawn from specified probability distributions. In practice, this conditional expectation is estimated through Monte Carlo sampling: for each fixed value of $Q_i$, multiple values of the remaining parameters are sampled, the model is evaluated for each parameter set, and the outputs are averaged. Repeating this process across the range of $Q_i$ values allows for estimating the variance of $E[Y \mid Q_i]$, which, when normalized by the total output variance, yields the first-order Sobol index. The first-order index thus provides a measure of how much $Q_i$ alone influences the overall variance in the model output.

**Total effect indices.** Letting the parameter of interest again be $Q_i$. The total effect Sobol sensitivity index $T_i$ represents the proportion of variance in the model output that is attributable to $Q_i$, including both its direct contribution and all

interactions with other parameters. It quantifies how much of the output variance would remain if all parameters except $Q_i$ were fixed. In other words, it measures the total influence of $Q_i$, encompassing its individual effect and its combined effects with other parameters. The index is defined as,

$$T_i = 1 - \frac{V[E[Y \mid Q_{\sim i}]]}{V[Y]},$$

(6)

where $Q_{\sim i}$ denotes all parameters except $Q_i$. Here, $E[Y \mid Q_{\sim i}]$ is the expected model output when all parameters other than $Q_i$ are fixed and $Q_i$ is allowed to vary. The variance $V[E[Y \mid Q_{\sim i}]]$ therefore reflects the remaining variability when $Q_i$ is the only source of uncertainty. Subtracting this from the total variance $V[Y]$ yields the total contribution of $Q_i$ to the output variance. The total effect index is especially useful for identifying parameters whose importance arises mainly through interactions, even if their individual (first-order) effects are small.

## Results

The results are split into two subsections: the results of the literature review that will determine the parameter range, and the results of the sensitivity analysis.

### Literature review

Within this section, we review the current literature split by parameter choice.

### Typical area scanned by flyers

Understanding the typical area scanned (denoted by $\mathcal{A}$ in the model) by female mosquitoes, or 'flyers', before finding an oviposition site is crucial for predicting mosquito dispersal patterns and assessing the spread of mosquito-borne diseases. The term 'flyer' in this context refers to a female mosquito that has consumed a blood meal and is searching for a suitable site to lay her eggs. While the primary focus is on the distance travelled by females seeking oviposition sites, various approaches have been employed to estimate the general dispersal distance (i.e., the distance that a mosquito moves from point of origin to where they are found later) of mosquitoes, including isotope labelling and genetic techniques, which provide baseline estimates for dispersal behavior.

The initial estimation in [15] for the typical area scanned by flyers was based on the linear regression analysis conducted by Diallo *et al.* (2011) [26]. They predicted that people and livestock could be protected if they lived more than 1500 m to the nearest ponds due to the linear regression suggesting that mosquitoes would not disperse further than that distance. Lo Iacono *et al.* [15] used this distance for the diameter of the area spanned, however, going forward we will use the recorded dispersals as the radius for the area scanned. In the sections that follow, we organise our review by publication to facilitate direct linkage to the original studies. For readers who prefer the information grouped taxonomically, the same data are reorganised by mosquito species in S1 Table of the Supporting Information.

Medeiros *et al.* (2017) [27] studied mosquito dispersal using isotope enrichment, revealing that 59% of *Culex quinquefasciatus* oviposition-site seeking females travelled between 1–2 km from their larval habitat. In contrast, 79% of oviposition-site seeking *Aedes albopictus* females stayed within 250 m. It should be noted that *Aedes albopictus* are container/treehole mosquito species, and not a flood pan/dambo species, so dispersal behavior may be different.

Ciota *et al.* (2012) [28] found that, among *Culex pipiens* and *Culex salinarius*, the mean distance travelled was 1.33 km. However, this study did not specifically focus on females searching for oviposition sites.

Linthicum (1985) [29] conducted an experiment in Kenya where a dambo was artificially flooded, leading to the emergence of *Aedes lineatopennis*. The mean dispersal distance for females of this species was 150 m.

Moore and Brown (2022) [30] reviewed the available data on the flight range of *Aedes aegypti* and concluded that the average flight distance for this species is 106 m. Another study of *Aedes aegypti* [31] found a mean dispersal distance of 45.2 m, with the distribution being Laplacian. This dispersal kernel (a probability distribution describing the likelihood of a mosquito being detected at various distances from its release point, regardless of the direction) was derived using a methodology that involved capturing adult female mosquitoes across high-rise apartment blocks with simple oviposition traps. The study reported that there is a 50% probability of effective dispersal occurring within 32 m and a 10% probability of flight extending beyond 100 m. This research used genetic techniques to identify closely related kin and the distances between them.

Verdonschot *et al.* (2014) [32] conducted a thorough review of mark-recapture experiments across the *Aedes* spp. genera and reported an average flight distance of 89 m for *Aedes* spp. mosquitoes and 609.5 m for *Culex* spp. mosquitoes. The review also highlighted the maximum flight distances: 2959 m for 24 species of *Aedes* spp. mosquitoes and 5014 m for 13 species of *Culex* spp. mosquitoes.

The values reported by Medeiros *et al.* (2017) [27] are particularly relevant for our analysis, as they specifically refer to oviposition-seeking *Culex* spp. females. The authors also provide a graph depicting the distribution of dispersal distances (see [27]), which aligns with a triangular distribution. Although the values are given in terms of radius, we can convert them to an area by calculating the area of a circle with the corresponding radius. For a triangular distribution, we require minimum, maximum, and mode values. Based on the data, the minimum area is 0.196 km² (radius 0.25 km), the maximum area is 15.904 km² (radius 2.25 km), and the mode area is approximately 9.621 km² (radius 1.75 km).

For *Aedes* spp. flyer flight range, based on the data available, we have selected a gamma distribution with shape 2.5 and scale 10000. See S1 Table for summary of results and S1 Appendix for the method to estimate this distribution.

## Proportion of water body area/soil in which eggs are laid

The proportion of the water body area or soil in which eggs are laid, denoted as $\kappa$ in the model, is an important parameter for understanding mosquito oviposition behavior. Soti et al (2012) [33] indicates that the inner distance from the pond border defining the laying area of *Culex* spp. on the water is about 1m. However, in general, there is a notable lack of specific empirical data on this parameter for both *Aedes* spp. and *Culex* spp. Consequently, due to the absence of concrete evidence, we apply a wide range of values for $\kappa$ spanning from 0.001 to 0.999, distributed uniformly, for both species. Here and throughout, a uniform distribution is used to represent a state of total ignorance within the chosen range, ensuring that no particular values are favoured. This approach avoids introducing bias into the model and allows all plausible parameter values to be explored equally in the sensitivity analysis.

## Number of eggs per batch

The oviposition strategies of *Aedes* spp. and *Culex* spp. mosquitoes differ significantly. *Culex* spp. mosquitoes typically lay their eggs in cohesive rafts on the water surface, whereas *Aedes* spp. mosquitoes deposit desiccation-resistant eggs singly on moist surfaces, allowing them to withstand dry conditions until favorable hatching conditions arise [34]. The number of eggs laid per batch, *b*, which are in general different, has been explored in various studies.

Several studies have quantified egg production within the *Culex* spp. genus. In *Culex pipiens molestus*, females deprived of a blood meal in the lab produced an average of 37 eggs per raft [35]. Another lab experiment found that blood-fed *Culex pipiens pallens* laid an average of 85.12±2.07 eggs per raft [36]. *Culex pipiens pipiens* captured from the wild produced significantly more eggs, averaging 213.9±8.7 per raft [37]. The number of eggs laid also varied with blood source: *Culex pipiens* fed on human, rabbit, and pigeon blood produced rafts containing 161.3±6.4, 102.2±5.8, and 146.1±4.7 eggs, respectively, with an overall range of 68–188 [38]. Similarly, *Culex pipiens* females fed on live quail blood produced an average of 199.89±7.24 eggs per raft [39], while *Culex quinquefasciatus* fed on chicken blood laid between 125 and 140 eggs per raft [40]. Age can also influence fecundity in *Culex quinquefasciatus*, with egg production per raft

ranging from 43 to 231 eggs across individuals of different ages. Although fecundity generally declined as mosquitoes aged, the mean number of eggs per raft remained high at 152.9 ± 43.4 eggs [41]. Sugar-deprived *Culex quinquefasciatus* females produced a similar number of eggs per raft (210 ± 12) compared to sugar-fed females (200 ± 10) [42], which is not particularly surprising since bloodmeals are more closely linked to fertility than sugar-feeding in mosquitoes.

In contrast, fewer studies have documented the egg production of *Aedes* spp. mosquitoes. Container-breeding species, such as *Aedes aegypti* and *Aedes albopictus*, typically lay eggs in artificial or natural containers such as water-filled tyres, pots, and tree holes, where water availability is relatively stable. In contrast, floodwater species like *Aedes mcintoshi* lay eggs in temporary water bodies formed by rainfall or flooding, which are highly transient and variable. These ecological differences influence their reproductive strategies and egg production patterns. However, available data allow for reasonable estimation of oviposition parameters. *Aedes aegypti* females fed on human, rabbit, and pigeon blood laid an average of 50.4 ± 2.7, 33.2 ± 1.8, and 46.3 ± 3.6 eggs per batch, respectively, with an overall range of 22–83 eggs [38]. Similarly, another study reported that *Aedes aegypti* females fed on rabbit blood produced 35–40 eggs per gonotrophic cycle [40]. Zeller and Koella [43] found that *Aedes aegypti* reared on a high amount of food throughout development laid the most eggs (67 ± 8.1), while those switched from high to low food produced the fewest (31 ± 5.2) (see [43]; Fig 3C). Both the likelihood of laying eggs and the total number of eggs were strongly influenced by the food available during late development. Padilha *et al.* [44] found that in the second and third gonotrophic cycles, *Aedes aegypti* females laid approximately 70 eggs per batch. Furthermore, Yan *et al.* [45] explored the effects of nutritional status, specifically larval and adult diet, such as nectar availability on fecundity. They reported that mosquito fecundity depends on both larval and adult nutrition: larval food quantity and adult food quality together determine egg production. Specifically, they found that well-nourished *Aedes aegypti* females laid an average of 90 eggs per batch, with a range of approximately 60–125 eggs.

Synthesizing this information, we propose that the number of eggs per batch in *Culex* spp. mosquitoes follows a normal distribution with a mean of 149 and a standard deviation of 16.2. In this choice, we include lab based experiments, as it may represent extreme case scenarios. For *Aedes* spp. mosquitoes, the best fit is a normal distribution with a mean of 55 and a standard deviation of 2.80, aligning with the observed fecundity ranges reported in the literature. See S1 Appendix for the method to estimate these distributions.

## Maximum surface density of eggs

Some mosquitoes avoid habitats that already have a high density of conspecific eggs or larvae. To represent this density-dependent competition, the model included a carrying-capacity term based on egg surface density, which limits how many eggs a water body can support. As eggs become crowded, oviposition rates fall. For this reason, egg density was included in the sensitivity analysis. Despite numerous studies on mosquito oviposition behavior, direct estimates of the maximum surface density of mosquito eggs $\rho$ (i.e., the number of eggs per unit area) were not returned from the scoping review. While some research has explored how female mosquitoes respond to existing eggs or larvae when selecting oviposition sites, these studies do not explicitly quantify maximum egg densities. Consequently, any estimates must be inferred from indirect evidence, and a broad parameter range is necessary to reflect this uncertainty.

For *Culex* spp. mosquitoes, an informative study by Shin *et al.* (2019) [46] investigated oviposition preferences in water containers with varying surface areas available for egg-laying. The authors noted that the highest number of egg rafts retrieved at once were 34, 164, and 32 for *Culex coronator*, *Culex nigripalpus*, and *Culex quinquefasciatus*, respectively, summing to a total of 230 egg rafts. Using our previously determined average of 150 eggs per raft, this yields an estimate of 34 500 eggs in a 0.06 m$^2$ (approximately a circular area of 14 cm radius) water body, equating to an approximate density of 575 000 eggs per m$^2$.

A second study investigated how the presence of varying numbers of pre-existing *Culex* spp. egg rafts influenced subsequent oviposition behavior [47]. The experiment included a range of egg raft quantities, with the highest treatment containing up to 100 egg rafts. Since converting these volumes to surface densities is challenging, and due to the absence of

definitive data on a biologically constrained maximum density, we adopt a conservative uniform distribution for this parameter, ranging from 500 000–20 000 000 eggs per m$^2$, based on findings from both studies.

For *Aedes* spp. mosquitoes, fewer studies provide insight into egg density. One study, aimed at improving *Aedes albopictus* ovitraps, recorded a maximum egg density of 0.49 eggs per cm$^2$, equivalent to 4.90 10$^3$ eggs per m$^2$ [48]. While this may not reflect the absolute maximum, it provides a useful lower bound for estimation. Another study investigating the distribution of *Aedes caspius* and *Aedes vexans* eggs among different vegetation types in Poland found that the highest egg densities, recorded in perennial nitrophilous plant communities, averaged 450.52±227.99 eggs per 400 cm$^2$ [49]. This translates to an average density of approximately 1.125 10$^4$ eggs per m$^2$, with an upper estimate (including standard error) of about 15 675 eggs per m$^2$.

Given the significant disparity between the estimated maximum densities for *Culex* and *Aedes* spp. mosquitoes, spanning several orders of magnitude, it is reasonable to expect that *Aedes* spp. egg density is inherently lower due to their oviposition strategy. Unlike *Culex* spp. mosquitoes, which lay tightly packed egg rafts, *Aedes* spp. deposit their eggs singly, often over dispersed surfaces [50]. However, the lack of direct studies quantifying maximum egg densities for *Aedes* spp. introduces considerable uncertainty. To account for this, we adopt a conservative approach by applying a broad uniform distribution, ranging from 12 000–1 000 000 eggs per m$^2$, ensuring that the full plausible range is considered.

### Livestock total and impact of the livestock on vector fecundity and gonotrophic cycles

Two additional parameters included in the analysis capture the influence of livestock on vector dynamics: the total number of livestock present and the impact of livestock on mosquito fecundity and gonotrophic cycle length, represented in the model as $q$. To our knowledge, there is no empirical literature quantifying the livestock–mosquito reproduction coefficient $q$. Both parameters have been assigned uniform distributions across their respective ranges, similar to that for the proportion of water body area/soil in which eggs are laid, representing a state of total ignorance and avoiding unjustified assumptions about their likely value.

### Parameter choice

Table 2 provides a comprehensive summary of the parameter ranges and distributions derived from the literature reviewed in the preceding sections. Each parameter, representing key aspects of mosquito dispersal and oviposition behavior, has

**Table 2. Parameter range and distribution summary.**

| Parameter | Symbol and Dimension | Distribution | Values | Rationale |
|---|---|---|---|---|
| Area scanned by *Culex* spp. before finding an oviposition site | $\mathcal{A}_C$ [L$^2$] | Triangular | Min = 196 000 m$^2$, Max = 15 904 000 m$^2$, Mode = 9 621 000 m$^2$ | [27] |
| Area scanned by *Aedes* spp. before finding an oviposition site | $\mathcal{A}_A$ [L$^2$] | Gamma | Shape = 2.5, Scale = 10 000 | [27,29–32] |
| Proportion of the water body area that eggs are laid on | $\kappa^{Culex}$ [-] | Uniform | Min = 0.001, Max = 0.999 | Arbitrary |
| Proportion of area on the soil where eggs are laid on | $\kappa^{Aedes}$ [-] | Uniform | Min = 0.001, Max = 0.999 | Arbitrary |
| Number of eggs laid per batch by *Culex* spp. | $b_C$ [-] | Normal | Mean = 149, SD = 16.2 | [35–42] |
| Number of eggs laid per batch by *Aedes* spp. | $b_A$ [-] | Normal | Mean = 55, SD = 2.80 | [38,40,44,45] |
| Eggs maximum density per m$^2$ | $\rho_C$ [L$^{-2}$] | Uniform | Min = 500000, Max = 20000000 | [46,47] |
| Eggs maximum density per m$^2$ | $\rho_A$ [L$^{-2}$] | Uniform | Min = 12000, Max = 1000000 | [48,49] |
| Parameter for the impact of the livestock on vector fecundity and gonotrophic cycles (or biting rate) | $q$ [-] | Uniform | Min = 1, Max = 10000000 | [48,49] |
| Livestock Total | livestocktotal [-] | Uniform | Min = 1, Max = 1000 | [48,49] |

been assigned an appropriate statistical distribution based on available empirical data or, where data is scarce, a conservative assumption. For dispersal area ($\mathcal{A}$), triangular and gamma distributions were chosen to reflect the observed variability in flight range. The proportion of water body or soil utilized for egg-laying ($\kappa$) is represented by a uniform distribution, due to the absence of direct evidence. The number of eggs per batch ($b$) follows a normal distribution, with species-specific means and standard deviations informed by multiple studies. Finally, the maximum surface density of eggs ($\rho$), the livestock total and the parameter which quantifies the impact of the livestock on vector fecundity and gonotrophic cycles ($q$) are modelled using a uniform distribution, reflecting the uncertainty in available estimates. This table consolidates all relevant parameter choices, ensuring a structured approach to model formulation.

### Sensitivity analysis

***Culex* spp. sensitivity analysis.** The overall results with fixed values of temperature, water body area, and detection probability all assigned to medium values (Fig 1), indicate that of the six parameters of interest, the most influential, in descending order, are the area scanned before finding an oviposition site ($\mathcal{A}_C$), the livestock–mosquito reproduction coefficients ($q$), the livestock total, the proportion of the water body area where *Culex* spp. lay their eggs ($\kappa^{Culex}$), the eggs maximum density per m² ($\rho_C$), and the number of eggs laid per batch ($b_C$).

Looking individually at temperature, water body area, and detection probability sheds some light on what causes a change in how influential the parameters are on the model output (Fig 2).

The analysis of the variation in indices between 21°$C$ and 29°$C$ reveals notable trends for both first-order indices and total effect indices. For the first-order indices of $\mathcal{A}_C$, there was a decrease of 15% between 21°$C$ and 25°$C$, followed by a smaller decline of 3% between 25°$C$ and 29°$C$. In terms of the total effect indices of $\mathcal{A}_C$, a 16% decrease was observed between 21°$C$ and 25°$C$, with a 3% decrease from 25°$C$ to 29°$C$. Additionally, increases can be seen across both livestock total and livestock–mosquito reproduction coefficient $q$, with a cumulative increase by 2.7 times and 3.1 times for first-order indices, respectively, and 1.2 times for total effect indices (for both). No significant changes were observed in the number of eggs laid per batch, eggs density or the proportion of area where the eggs are laid.

The analysis of the variation in indices between 10 000 m² and 25 000 m² reveals notable trends for both first-order indices and total effect indices. As the water body area increases, the relative importance of livestock total and livestock–mosquito reproduction coefficient $q$ increases to approximately 22.2 times and 17.9 times their original values, respectively, for first-order indices, and to about 1.9 times and 1.6 times their original values, respectively, for total effect indices. Notably, the area scanned decreases in importance as the water body increases, dropping to about one-half of its original

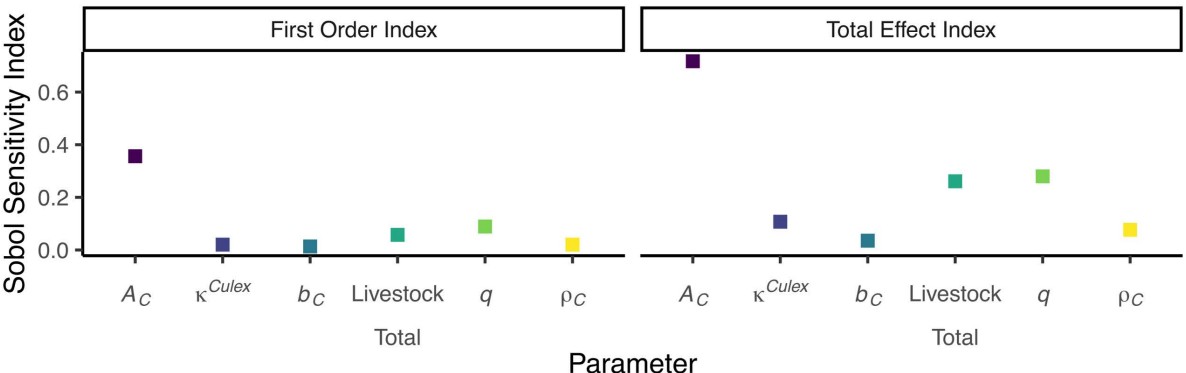

**Fig 1. First order and total effect Sobol sensitivity indices for *Culex* spp. mosquitoes.** With medium values of temperature, water body area and detection probability.

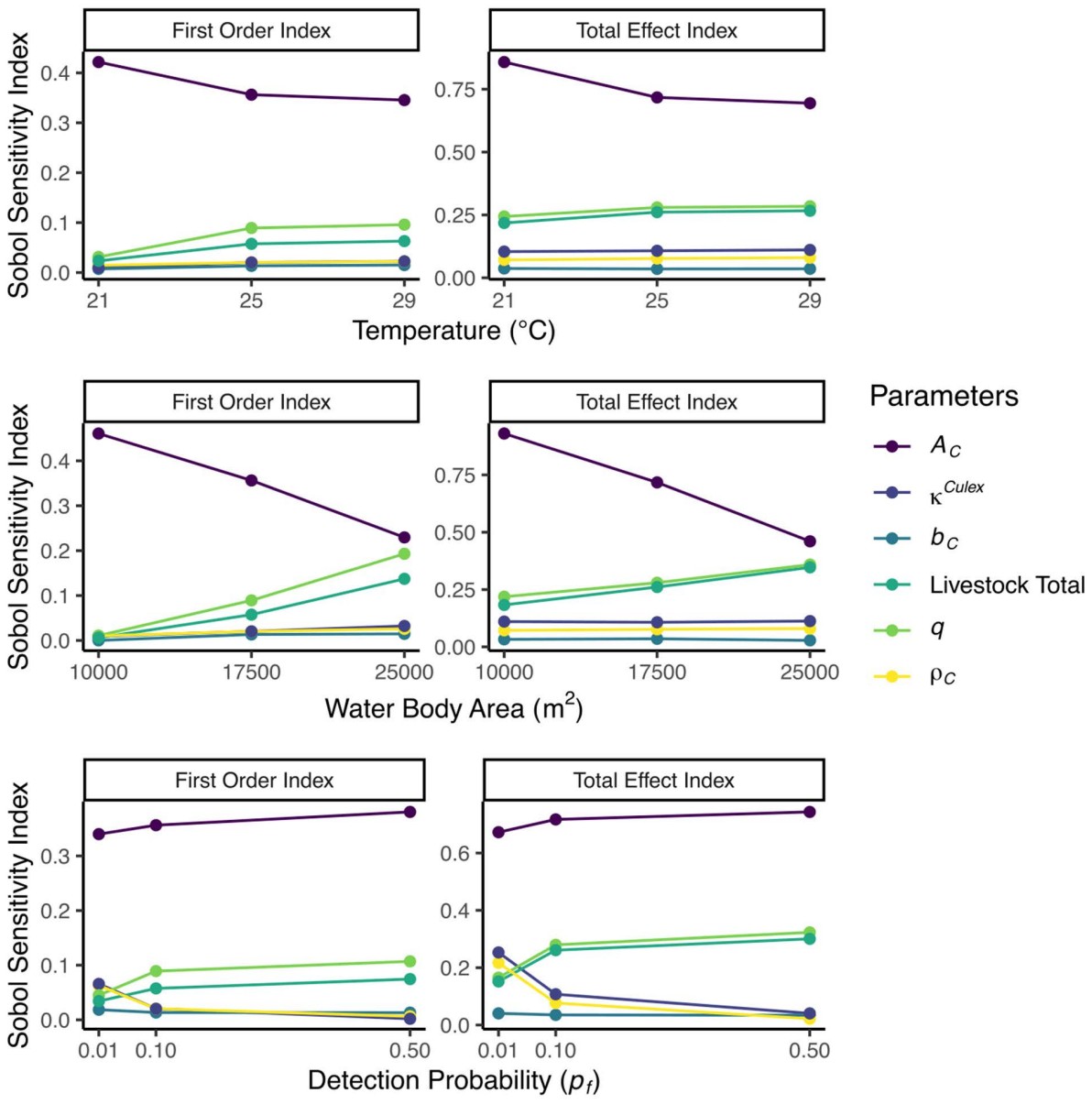

**Fig 2. Sobol sensitivity indices for *Culex* spp. mosquitoes.** Comparing varying values of constant: temperature, water body areas, and detection probabilities.

value for both first-order and total effect indices, with its importance notably reduced when the water body area reaches 25 000 m². Further, there are no notable changes in the number of eggs laid per batch, eggs density or the proportion of area where the eggs are laid.

As the value of detection probability increases, we see an increase in the area scanned, livestock total and the livestock–mosquito reproduction coefficient $q$. Most notable is the change from 0.01 to 0.1 for both livestock total and $q$, of about 1.7 times. However, a decrease can be observed in both the egg's density and the proportion of area where the eggs are laid.

***Aedes* spp. sensitivity analysis.** The same set of analyses were repeated for the equivalent *Aedes* spp. model. The overall results, where medium values of temperature, water body area, and detection probability were used, show that, the proportion of water body where eggs are laid ($\kappa^{Aedes}$) was the most influential parameter (Fig 3). The remaining five parameters had a different order of importance to that seen in the *Culex* spp. analysis, with the second most influential parameter being $\rho_A$, the third being $q$, followed by $\mathcal{A}_A$, livestock total, and $b_A$.

Looking individually at temperature, water body area, and detection probability reveals no notable trends for both first-order indices and total effect indices (S2 Fig). The order of importance and the index remains relatively constant throughout.

**Periodic temperature and water body.** To capture temporal dynamics, we perform a time-varying Sobol analysis using the periodic functions for temperature (Eq. (3)) and water bodies (Eq. (4)). When the periodic functions are in-sync (i.e., the peaks of the temperature and mean surface area of water bodies are observed at the same time), the results for *Culex* spp. are presented in Fig 4 and *Aedes* spp. in Fig 5. When they are out-of-sync, the results can be observed in S3 Fig. The plots highlight the changing of the Sobol sensitivity indices for each of the six parameters of interest, alongside the plot for the mean temperature (in Kelvin), mean surface area of water bodies (in $m^2$) and the average mosquito population across all the runs.

It can be highlighted for the in-sync figures, from the panels on the right, that there is a clear periodic pattern in which the peaks in the mosquito population align with the periodic behaviour of both the temperature and water bodies. For *Culex* spp., it can be observed that the area scanned remains the highest sensitivity index, exhibiting pronounced fluctuations, with its minimum occurring approximately 50 days after the peaks in both temperature and water availability. In contrast, livestock total displays a similarly periodic pattern but reaches its maximum when the area scanned is at its minimum, indicating an inverse relationship between these two drivers. Meanwhile, the parameters $\kappa^{Culex}$, $b_C$, and $\rho_C$ show very limited variation across the simulation period, suggesting that they contribute comparatively little to the overall temporal dynamics of the population.

On the other hand, for *Aedes* spp., the patterns differ noticeably to the periodic *Culex* spp. results, but not far dissimilar to the constant conditions for *Aedes* spp. The parameters $A_A$ and $b_A$ remain effectively unchanged throughout, showing minimal sensitivity to the population cycle. In addition, $\kappa^{Aedes}$ and $\rho_A$ closely mirror one another, both exhibiting a distinct dip when the *Aedes* spp. population reaches its lowest point. In contrast, livestock total and livestock–mosquito reproduction coefficient $q$ remain largely constant across the cycle but show a marked increase precisely when the population is at its minimum, revealing an inverse response relative to the parameters that decline.

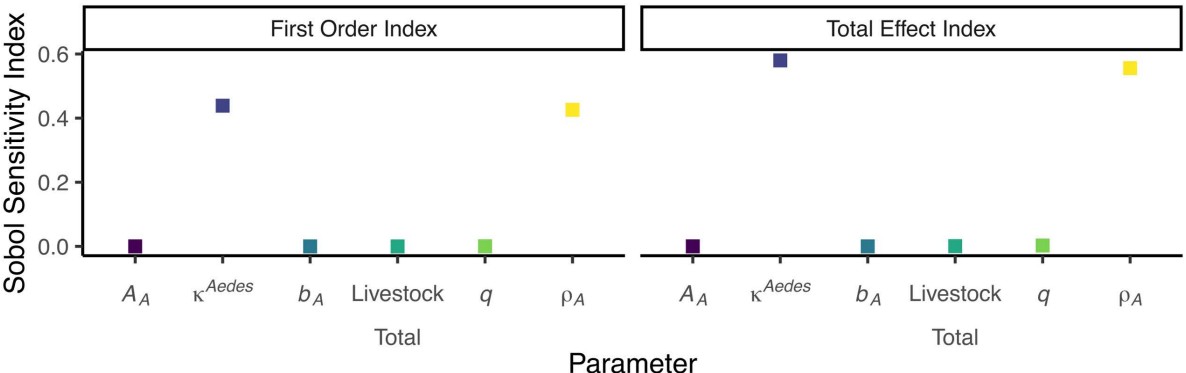

**Fig 3. First order and total effect Sobol sensitivity indices for *Aedes* spp. mosquitoes.** With medium values of temperature, water body area and detection probability.

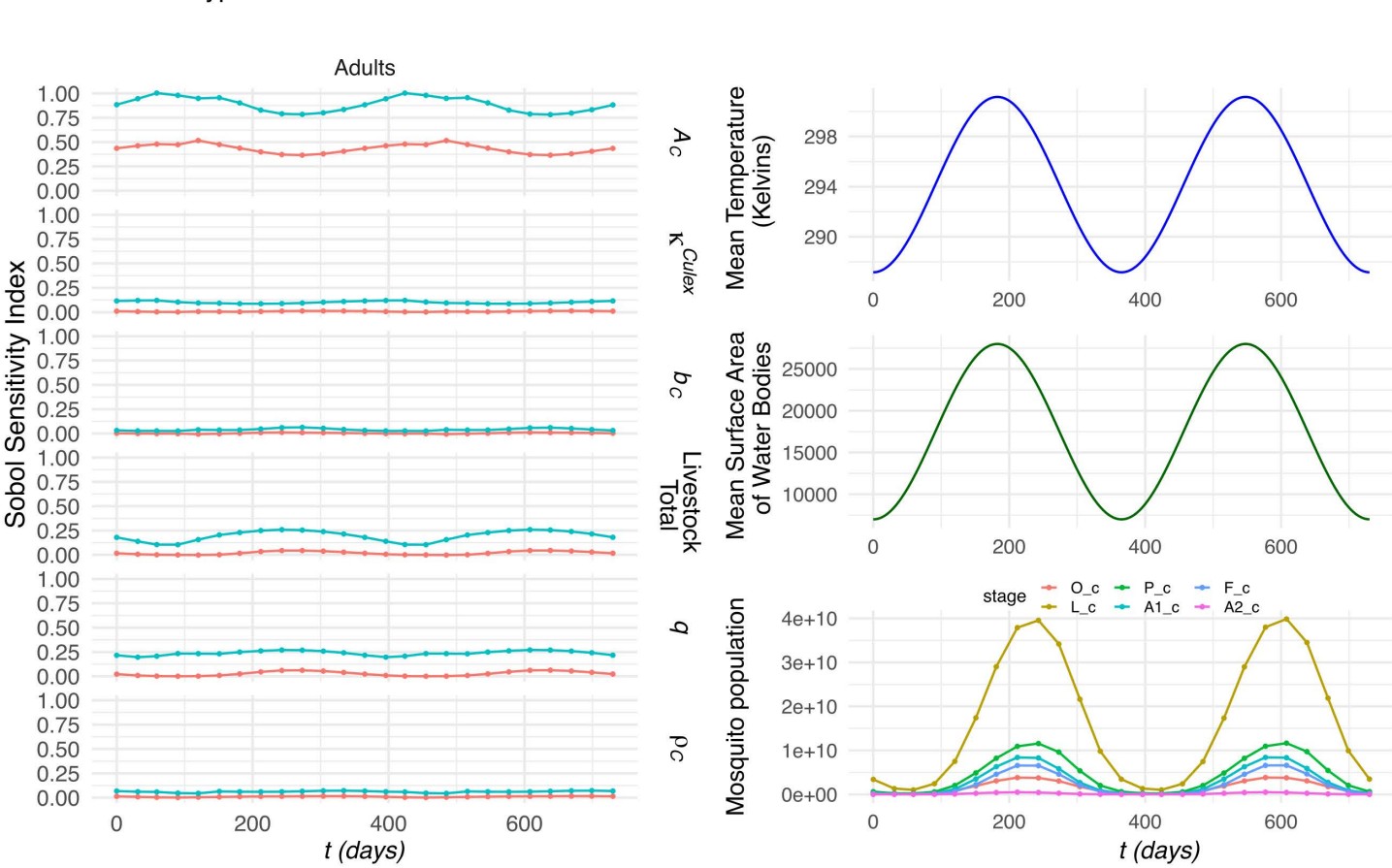

**Fig 4. Sobol sensitivity in-sync time-varying indices for *Culex* spp.** Comparing the fluctuations in mean temperature and mean surface area of water bodies, with the mean mosquito population and the Sobol indices over time. The stages for the *Culex* spp. mosquito population is divided into six stages: eggs (O_c), larvae (L_c), pupae (P_c), nulliparous females (A1_c), flyers (F_c), and parous females (A2_c).

When comparing to out-of-sync results (S3 Fig), it can be observed there is minimal variation in the parameters for the *Culex* spp. system, matching the constant parameters. Yet, when observing the results for *Aedes* spp., we see the same pattern as Fig 5, but we see the peaks and minimums of the indices in line with the minimum of the water body areas.

## Discussion and conclusion

This study aimed to clarify the ecological drivers of mosquito population dynamics that underpin the transmission of Rift Valley fever virus (RVFV), with a particular focus on vector ecology. A deterministic, compartmental ecoepidemiological model designed for RVFV transmission was used to capture the interactions between mosquito life cycles, environmental conditions, and disease spread. Although the underlying model from [15] is capable of simulating infection and pathogen spread, the Sobol sensitivity analysis was performed on the ecological (non-infectious) component only. Consequently, disease spread via vector dispersal was not assessed, and movement of infected livestock was not included. However, several key ecological parameters in the model, such as those governing oviposition and the carrying capacity, remain poorly constrained by empirical data, introducing uncertainty into model predictions. To address this, we conducted a scoping review followed by a sensitivity analysis to quantify the relative influence of these uncertain parameters on

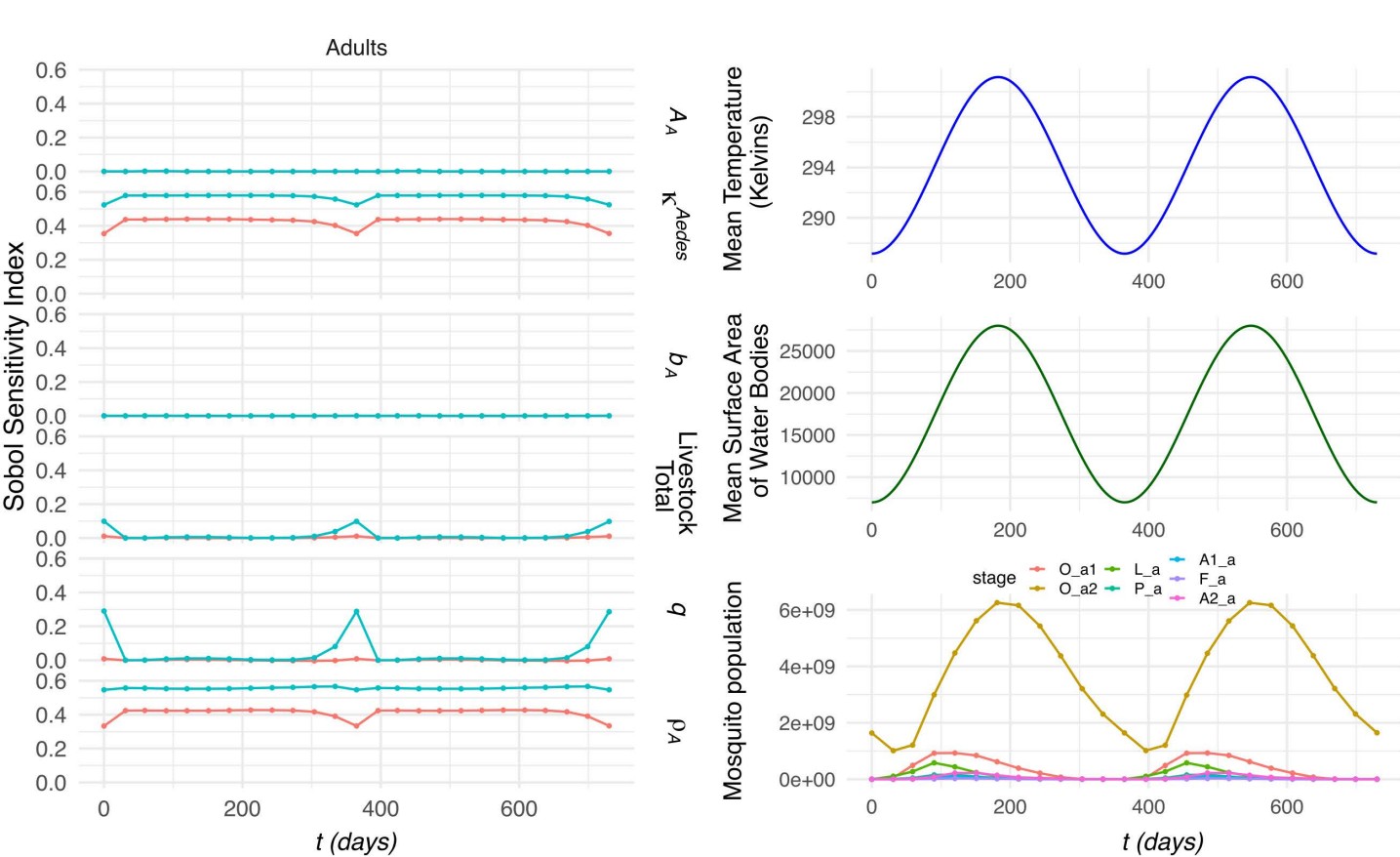

**Fig 5. Sobol sensitivity in-sync time-varying indices for *Aedes* spp.** Comparing the fluctuations in mean temperature and mean surface area of water bodies, with the mean mosquito population and the Sobol indices over time. The stages for the *Aedes* spp. mosquito population is divided into seven stages: immature eggs (O_a1), mature eggs (O_a2), larvae (L_a), pupae (P_a), nulliparous females (A1_a), flyers (F_a), and parous females (A2_a).

mosquito abundance. By identifying gaps in data collections and determining which parameters most strongly affect model outcomes, our goal is to guide future data collection efforts, improve the reliability of model-based assessments and better inform RVFV risk and control strategies.

## Gaps in data collections

The scoping review highlights a scarcity of research in some areas. For instance, while the literature contains numerous references to mosquito egg production, much of this research is laboratory-based and often features experimental variables being tested (such as diet) that make the numbers difficult to generalize to field scale. Furthermore, the numbers also vary considerably within and between species (see Table 2). We found far fewer articles referring to egg production of *Aedes* spp. mosquitoes than *Culex* spp. mosquitoes. Furthermore, all studies on the genus *Aedes* spp. identified in the literature focused on a single species, *Aedes aegypti*, highlighting the paucity of species-specific data for other genera.

Our findings highlight a significant knowledge gap in understanding the spatial patterns of mosquito oviposition across larger water bodies. Most dispersal estimates reported in the literature are derived from mark–release–recapture studies

of host-seeking or newly emerged adults, rather than gravid, oviposition-seeking females (see, e.g., [51,52]). Because gravid females may exhibit skip-oviposition or site fidelity depending on species and habitat availability, we consider dispersal to be uncertain and therefore adopt a conservative approach.

In addition, we found a similar dearth of studies observing the density of mosquito eggs in the field (or, indeed, in the laboratory). Most significantly, based on our scoping review, we note an absence of research into spatial oviposition dynamics across water bodies. For *Aedes* spp. mosquitoes, our particular measure of interest refers to the area of the ring of damp soil around the water body in which eggs are laid, divided by the area of the water body itself, whereas for *Culex* spp. mosquitoes this refers to the actual surface area of water across which eggs are laid divided by the area of the whole water body. While we could not gather any information from the literature to influence our specific parameter estimates, it is important to state that we were unable to find any data regarding patterns of mosquito egg laying across water bodies.

### Identifying influential parameters

The sensitivity analysis showed that the area scanned by *Culex* spp. is the most sensitive, in contrast to the situation for *Aedes* spp. which exhibits a very low Sobol sensitivity index. This is irrespective of the temperature, probability of detection and in part for water bodies. However, the range of scanned-area values is much wider for *Culex* spp. than for *Aedes* spp., so the large Sobol index may simply reflect the greater variance in this input parameter. It is also noteworthy that, when comparing the maximum proportion of scanned area that is water (using the median water-body area), about 17% of the area scanned by *Aedes* spp. is water, compared with only 0.11% for *Culex* spp.

The analysis also showed that in both populations the indexes for the eggs maximum density and the proportion of area of the water body/soil where mosquitoes lay their eggs have similar value. This is expected because both parameters only affect the carrying capacity for eggs and the minimal differences are expected due to only the ranges of the chosen values.

For both models, the number of eggs laid is among the least sensitive parameters. The ranges observed in the literature, 106.7 eggs for *Culex* spp. and 18.5 eggs for *Aedes* spp., reflect differences due to species, feeding status, and experimental conditions. However, the sensitivity analysis suggests that even across this biologically realistic range, the impact on the overall model output is minimal. This indicates that variability in egg production due to factors such as whether mosquitoes are blood-fed or the specific species used is unlikely to significantly influence model predictions. Therefore, any value within the reported range can be considered suitable for use in the model without substantially affecting its robustness.

According to the constant sensitivity analysis, in both populations, changing the value of the probability of a mosquito detecting livestock ($p_f$) has little to no impact on the relative importance of the indices. In the current model with no disease, this probability merely rescales the vector-to-host ratio by a constant factor. The vector-to-host ratio, in turn, influences the density-dependent number of eggs per batch and the biting rate (Eq. (6), (10), (13), and (20) in [15] based on [17]). However, this effect is negligible under the published model, which assumes that host density only becomes impactful when the number of hosts is extremely low compared to the number of vectors (mathematically, this is because the parameter $q = 10^{11}$ in [15] approaches infinity, where q represents the vector-to-host ratio at which vector fecundity is halved). This factor is expected to become important however, when the low number of hosts limits blood meal significantly reducing the number of eggs per batch and the biting rate. The vector-to-host ratio is also expected to be important in an epidemiological situation as this affects the force of infection. Hence, since the livestock–mosquito reproduction coefficient q is not explicitly defined or explored in the existing literature, this motivated its inclusion in the Sobol sensitivity analysis to assess its potential influence when treated as a finite, biologically interpretable parameter, which from the results is an important parameter to investigate for *Culex* spp. and investigate for *Aedes* spp.

When considering the changes in temperature in the *Culex* spp. population (Fig 2), the results suggest that the indices are more sensitive to temperature changes between 21°C and 25°C, with a slower decline observed beyond 25°C. This

effect is less evident from the time-dependent periodic analysis, with the relatively smooth fluctuations of indices. Since temperature is incorporated into the development rates and mortality rates (see S3 Appendix), then it is not obvious how temperature influences the changes highlighted in the observations.

On the other hand, when considering the water body area, the parameters do not uniformly become more or less sensitive to changes. It is observed that the area scanned, $\mathcal{A}_C$, loses sensitivity, while others parameters (livestock total and the parameter for the livestock–mosquito reproduction coefficient $q$) gain it. A possible explanation for this is the structure of the equations themselves: as seen in [15] supplementary information, Eq. (3), the influence of $\mathcal{A}_C$ is relative to the oviposition rate, so an increase in water surface area diminishes the effect of $\mathcal{A}_C$. Hence, we observe a loss in sensitivity. Meanwhile, Eq. (5) shows that increases in water body area directly amplifies the carrying capacity (with the equation becoming more sensitive to changes in $\rho$ or $\kappa$), which then feeds into Eq. (4), increasing the variance in the egg load rate. Therefore, as water body area expands, parameters like $\rho$ and $\kappa$ exert a stronger influence on the system. Livestock abundance and the livestock–mosquito reproduction coefficient $q$ influence vector fecundity (Eqs. (6) and (10) in [15] SI) and gonotrophic cycles (Eqs. (13) and (20)), these also depend on temperature and water body area. These parameters are somehow expected to show greater sensitivity (larger variances) under higher temperatures and larger water bodies; e.g., in the extreme case of no water body, fecundity is zero and livestock abundance and livestock–mosquito reproduction coefficient $q$ have no effect.

In comparison, when studying the *Aedes* spp. model, we find the most sensitive parameters throughout the analysis are the proportion of area of the soil where mosquitoes lay their eggs on ($\kappa^{Aedes}$) and the maximum density of eggs on the soil ($\rho_A$). This is likely due to the interaction of the two parameters in Eq. (9) [15]. Yet, in this analysis, we did not see changes in any other parameter when we changed the temperature, water body area or the detection probability.

These first- and total-order results under different scenarios clarify which parameters exert the strongest direct influence. To determine whether interactions among parameters add further structure to the model response, we also considered the second-order indices (S4 Appendix). From the results, the second-order contributions were comparatively small, reinforcing that the dominant sources of uncertainty are driven by individual parameter effects rather than synergistic interactions. The few interaction pairs that did emerge, however, help clarify where model dynamics may be sensitive to combinations of behavioural or spatial parameters. For *Culex* spp., when the climate parameters are set to the medium values (matching Fig 1), the strongest interaction pairs are the area scanned with livestock–mosquito reproduction coefficient $q$ or livestock total. Their appearance as the strongest interaction pairs suggests a weak but consistent coupling; this is perhaps not surprising since host and feeding-related processes act together on the gonotrophic cycles and fecundity. However, the magnitudes of the indices (<0.1) indicate that these interactions are unlikely to significantly influence overall model uncertainty under the parameter ranges considered. Further, when considering the pair for the *Aedes* spp. analysis of second-order indices, the emergence of the pair $\rho_A$ and $\kappa^{Aedes}$ as the only meaningful interaction is unsurprising. The first- and total-order Sobol indices showed that the maximum egg density and the proportion of suitable oviposition area were the only parameters exerting substantial individual influence on the model output. Their coupling in the second-order analysis therefore reinforces the central role these two processes play in shaping *Aedes* spp. population dynamics, both independently and in combination. Overall, the small second-order indices for both mosquito genera indicate that parameter effects are largely independent, with only weak non-linear interactions, suggesting that most variability can be explained by individual parameters.

To better understand the source of the observed sensitivity patterns, we conducted an additional analysis using only the four ecological parameters (keeping livestock total and livestock–mosquito reproduction coefficient $q$ constant) in which the Sobol sampling matrices generated for one mosquito system were applied to the other (see S5 Appendix). Specifically, the matrices created for the *Aedes* spp. model were used in the *Culex* spp. model, and vice versa. We found that each system returned sensitivity indices that closely resembled those from the original model associated with the Sobol matrix. For example, applying the *Aedes* spp. Sobol matrices to the *Culex* spp. model produced results similar to those obtained

in the *Aedes* spp. model. This indicates that the sensitivity patterns are primarily driven by the parameter ranges and distributions used in the Sobol sampling, rather than by differences in the mosquito system structure itself.

Extending the analysis, we incorporated fluctuating temperature and water body values using periodic functions (following the approach in [15]) and using time-varying Sobol sensitivity analysis to capture the changes during these fluctuations. The indexes exhibit weak seasonality, reflecting changes in mosquito abundance. For *Culex* spp., Sobol sensitivity indexes for total livestock and the livestock–mosquito reproduction coefficient ($q$) increase with higher mosquito populations, whereas *Aedes* spp. shows the opposite trend. This contrast likely arises because these parameters influence biting rate and the gonotrophic cycle through the inverse of their value scaled by mosquito abundance. At low abundance, as in *Aedes* spp. here compared to *Culex* spp., the relationship is steep, producing high sensitivity at small values and flattening at larger values. At high abundance, sensitivity stabilizes at intermediate levels across values of total livestock and $q$. Thus, sensitivity is not a simple monotonic function of mosquito abundance, explaining the divergent patterns observed between *Aedes* spp. and *Culex* spp. Apart this aspect, the sensitivity results reveal that the order of the Sobol indices remains consistent with those obtained under constant medium values. This consistency suggests that the model's behavior is stable under both static and dynamic environmental conditions, and further confirms that the relative influence of each parameter is robust and not significantly altered by environmental variability.

## Implication for public health and control strategies

Although the approach is based on a model developed for RVFV transmission, its findings are relevant to other mosquito species with oviposition behaviours similar to those of *Aedes* spp. and *Culex* spp., which exemplify egg-laying on water and on moist soil, respectively.

To our knowledge, this is one of the first studies to apply global sensitivity analysis to mosquito ecology. The current analysis identified the input parameters that most strongly influence the uncertainty and behaviour of model outputs. This is particularly important for complex, non-linear models in which interactions among inputs can substantially affect outcomes. It is important to note that a large Sobol index may sometimes indicate a parameter's high uncertainty rather than a strong intrinsic influence on model behaviour. This uncertainty is epistemic in nature, as it arises from limited field data, and could be reduced through targeted data collection. However, it is important to acknowledge the challenges of directly measuring some mosquito ecological and biological parameters, such as lifespan, due to practical and logistical constraints. In such cases, indirect estimation methods, modeling approaches, or sensitivity analyses may provide valuable insights to address these data gaps. When such a parameter is fixed or better characterised, the total output variance is reduced in proportion to its Sobol index. Therefore, reducing data gaps and improving parameter estimation are essential to ensure that variability reflects genuine biophysical differences rather than knowledge uncertainty. Regarding public health implications, the large Sobol index for the area scanned by *Culex* spp. females before oviposition may partly reflect knowledge uncertainty. Nevertheless, because this parameter reflects the spatial distribution of breeding sites, and a high sensitivity suggests that source-reduction measures, such as removing standing water, improving drainage, and maintaining water-management infrastructure, could substantially reduce mosquito populations. The time-varying analysis further shows that its total Sobol index peaks as adult numbers begin to rise, indicating that source reduction is most effective during this period. However, interventions near case locations may inadvertently increase dispersal of potentially infected vectors (see [53]), an effect not represented in the current model. For *Aedes* spp., the area scanned before oviposition has a low Sobol index due to the narrower range of values reported in the literature. We cannot, however, rule out that these values may vary more widely in settings not represented in existing studies. The proportion of water-body area or soil used for oviposition is also influential, particularly for *Aedes* spp., reinforcing the message that disrupting breeding sites, such as draining stagnant water, improving soil drainage, or filling low-lying areas to reduce breeding sites, may be especially effective for mosquito control.

In the current model, the maximum egg density per unit area primarily determines the system's carrying capacity for egg deposition. Since these parameters are unlikely to be directly affected by control strategies, they are not expected to have immediate public health implications. Adding further complexity, the relationship between larval density and the number of adults is not straightforward, with overcompensation observed in several mosquito species (see, for example, [54–56]). Nonetheless, higher egg densities can result in dense larval populations, where resource competition influences survival rates, an effect not accounted for in the current model.

The time-varying Sobol analysis indicated that the number of eggs laid per batch by adult females had a greater influence on modelled mosquito population size at the time when egg numbers began to increase rapidly, suggesting that egg-killing interventions, and possibly larvicide application, are most effective during this period, as expected. The impact of direct egg elimination appears to be of secondary importance compared to measures targeting other traits, such as source reduction that hampers gravid female dispersal. However, firm conclusions can only be drawn once the uncertainty in the input parameters is reduced to reflect true biophysical variability rather than knowledge gaps.

## Limitations and future research directions

Inevitably, the present approach has some limitations. Since the model is set up to read empirical data for air temperature, water body area and livestock numbers, we initially chose to set these as constants to allow for a more transparent analysis. This approach helps isolate the influence of other parameters and ensures that any observed sensitivity in the Sobol analysis is not confounded by fluctuations in these environmental inputs. However, a major limitation for this is that the model for *Aedes* spp. relies on the fluctuation of the water body for the hatching of the mature eggs to larvae, an important mechanism that is suppressed when water availability is fixed. To address this, we also repeated the analysis using periodically varying temperature and water body area, based on Eq. (3)-(4), which more realistically capture seasonal dynamics and enable the emergence of key temporal patterns in mosquito population behavior. Nevertheless, the analysis was firstly conducted with the three levels of constant temperature, water body area, and also the probability of a mosquito detecting livestock ($p_f$ in the model). Regarding this parameter, it should be noted that, by construction, the density-dependent model for the number of eggs per batch (Eq (6) and (10) in the Supplementary Information of [15]) is only valid when the vector-to-host ratio, and therefore the parameter $p_f$, is strictly greater than zero.

We chose to utilize Sobol sequences to generate our parameter sample sets due to their proven advantages in sensitivity analyses. Compared to Latin Hypercube Sampling (LHS) and random sampling, Sobol sequences have been shown to result in faster convergence, lower computational cost, and improved reproducibility [57]. This method ensures that the parameter space is sampled more uniformly, which is critical for reliable sensitivity analysis.

Given the model's robustness under both constant and fluctuating environmental conditions, efforts should now focus on refining key ecological assumptions related to oviposition. Priority areas include quantifying the proportion of water-body or soil area actually used for egg laying and the typical area scanned by flying *Culex* spp. mosquitoes when seeking sites. Addressing these gaps will improve the spatial realism of the model and strengthen predictions of mosquito population dynamics in heterogeneous or resource-limited environments.

Additionally, approximately 40 mosquito species [58] and multiple vertebrate hosts, including wildlife, have been implicated in the transmission of RVFV [59]. In this study, we did not account for heterogeneity in host abundance or for interspecific differences in mosquito feeding preferences. Such behavioural variation can substantially influence mosquito ecology and the dynamics of vector-borne diseases [60]. Nevertheless, many mosquito species are considered opportunistic feeders and are likely to bite a range of mammalian hosts. Future work should therefore investigate in greater detail how biodiversity and host composition shape mosquito behaviour and ecology and RVFV transmission dynamics.

The analysis was conducted for a typical East African setting, where RVFV transmission is largely driven by *Aedes* spp. and *Culex* spp., and their dynamics are closely linked to dambo ecology. This might contrast with the epidemiology observed in other geographic regions that lack, for instance, the periodic dambo systems characteristic of East Africa.

We included the effect of livestock density, as a blood source, on egg batch size and the gonotrophic cycle, assuming egg production requires blood meals. The relationship between host abundance and these traits was adapted from studies on Chagas disease vectors for triatomines [17]. Experimental work on *Culex quinquefasciatus* [61] showed higher fecundity and fertility in chicken-fed than in mouse-fed mosquitoes, with seasonal variation only in the latter. However, quantitative data linking host abundance to mosquito reproductive traits remain scarce. Further studies are needed to clarify how host density, distribution, and diversity influence reproductive and life-history traits such as egg batch size, gonotrophic cycle, survival, and fertility.

In addition, seasonal variation in temperature and water bodies was represented using two periodic functions. While this provides a simplified approximation of seasonality, future analyses should consider varying phase and amplitude, as well as the influence of extreme weather events, which are not captured by these representations but may become increasingly important under changing climatic conditions.

Here, we used small water bodies as proxies for breeding sites because their use eliminate the temporal gap between rainfall and site formation and for RVFV, dambos are the most relevant breeding habitats. However, mosquitoes also breed in containers, gutters, fields, urban infrastructure, and flood-prone areas, for which rainfall may serve as a more suitable proxy.

The model is non-spatial and does not explicitly incorporate the effects of vegetation or predation on larval or adult mosquito stages. A simple circular geometry was assumed for breeding sites, and the effects of terrain, spatial fragmentation, and other landscape heterogeneities were not represented. Including such spatial complexity in future work would improve the realism of predictions and support the development of more targeted vector control strategies. Further, this could be envisioned once more biological data are available, such as the ones shown to be critical in this manuscript, to reduce the complexity and the actual necessary approximations.

Further research is needed to clarify how egg abundance relates to the area and perimeter of water bodies, and whether risk is higher in ponds with smooth or jagged edges. For parsimony, the current model uses the total area of water bodies within a location and does not account for landscape fragmentation. Multiple smaller water bodies may therefore present a higher risk of egg deposition than a single water body of equivalent total area. Improved estimation of these parameters will enhance model accuracy and strengthen risk assessments.

Conducting sensitivity analysis will help refine the model, making it more efficient and guiding its calibration. Once optimized, the model can be used to investigate important control strategies, such as disrupting female access to breeding sites, assessing the effectiveness of mass larviciding, and determining the optimal timing and locations for larviciding applications in dambo systems to maximize impact.

This study represents an important step toward model-guided fieldwork by identifying the parameters that most require accurate measurement, either because they strongly influence mosquito ecology or because their uncertainty limits model predictive power. More broadly, progress on RVFV and other vector-borne diseases is constrained by limited integration across disciplines. Improving prediction of vector ecology and disease dynamics requires stronger links between modelling, ecological and epidemiological fieldwork, and community engagement. Such integration is essential for models to reflect ecological complexity and remain relevant to public health needs. Strengthening these connections will improve our ability to anticipate changes in vector populations and disease risk, and to rigorously assess context-specific interventions.

## Supporting information

**S1 Table. Summary of literature search for the typical area scanned by flyers.** (PDF)

**S1 Fig. Sobol sensitivity convergence analysis.** Evaluating the robustness of the Sobol sensitivity indices with increasing sample sizes. (PDF)

**S2 Fig. Sobol sensitivity indices for *Aedes* spp. Comparing varying values of constant: temperature, water body areas, and detection probabilities.**
(PDF)

**S3 Fig. Time varying Sobol sensitivity indices for *Culex* spp. and *Aedes* spp. with out-phase periodic functions for temperature and water bodies.**
(PDF)

**S1 Appendix. Calculation of Distributions.**
(PDF)

**S2 Appendix. Parameters.** Input parameters for the model.
(PDF)

**S3 Appendix. Investigation into parameters.** Investigating the impact of development rate, mortality rate and oviposition rate when there is constant temperature and water body.
(PDF)

**S4 Appendix. Investigation into second-order indices.** Extending the analysis for the constant scenario to investigate higher order Sobol indices.
(PDF)

**S5 Appendix. Investigation into differences.** Using the same Sobol matrices, we put them in the opposite mosquito system to observe any differences.
(PDF)

## Acknowledgments

For the purpose of Open Access, the author has applied a Creative Commons Attribution (CC BY) public copyright licence to any Author Accepted Manuscript version arising from this submission.

## Author contributions

**Conceptualization:** Giovanni Lo Iacono.

**Data curation:** Jessica Furber, Sophie North.

**Formal analysis:** Jessica Furber, Sophie North.

**Funding acquisition:** Giovanni Lo Iacono.

**Investigation:** Jessica Furber, Sophie North.

**Methodology:** Giovanni Lo Iacono.

**Project administration:** Jessica Furber.

**Resources:** Giovanni Lo Iacono.

**Software:** Jessica Furber, Sophie North, Giovanni Lo Iacono.

**Supervision:** Martha Betson, Christophe Boëte, Giovanni Lo Iacono.

**Validation:** Martha Betson, Christophe Boëte, Giovanni Lo Iacono.

**Visualization:** Jessica Furber, Daniel Horton.

**Writing – original draft:** Jessica Furber, Sophie North.

**Writing – review & editing:** Jessica Furber, Martha Betson, Christophe Boëte, Daniel Horton, Giovanni Lo Iacono.

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
