## [Decision Letter · Decision Letter 0]

23 Sep 2025

Sensitivity analysis of factors influencing the ecology of mosquitoes involved in the transmission of Rift Valley fever virus

Dear Dr. BOETE,

Thank you for submitting your manuscript to PLOS Neglected Tropical Diseases. After careful consideration, we feel that it has merit but does not fully meet PLOS Neglected Tropical Diseases's publication criteria as it currently stands. Therefore, we invite you to submit a revised version of the manuscript that addresses the points raised during the review process.

Please submit your revised manuscript within 60 days Nov 22 2025 11:59PM. If you will need more time than this to complete your revisions, please reply to this message or contact the journal office at plosntds@plos.org. Please include the following items when submitting your revised manuscript:

We look forward to receiving your revised manuscript.

Kind regards,

Ran Wang, M.D.

Academic Editor

Audrey Lenhart

Section Editor

Shaden Kamhawi

co-Editor-in-Chief

Paul Brindley

co-Editor-in-Chief

**Additional Editor Comments:**

Reviewer #1:

Reviewer #2:

Reviewer #3:

Reviewer #4:

**Journal Requirements:**

At this stage, the following Authors/Authors require contributions: Christophe Boëte. Please ensure that the full contributions of each author are acknowledged in the "Add/Edit/Remove Authors" section of our submission form.

4) Please amend your detailed Financial Disclosure statement. This is published with the article. It must therefore be completed in full sentences and contain the exact wording you wish to be published.

5)  Please ensure that the funders and grant numbers match between the Financial Disclosure field and the Funding Information tab in your submission form. Note that the funders must be provided in the same order in both places as well.

**Reviewers' Comments:**

Reviewer's Responses to Questions

**Key Review Criteria Required for Acceptance?**

**Methods**

-Are the objectives of the study clearly articulated with a clear testable hypothesis stated?

-Is the study design appropriate to address the stated objectives?

-Is the population clearly described and appropriate for the hypothesis being tested?

-Is the sample size sufficient to ensure adequate power to address the hypothesis being tested?

-Were correct statistical analysis used to support conclusions?

-Are there concerns about ethical or regulatory requirements being met?

Reviewer #1: The methods are appropriately described and the sensitivity analysis is interesting. In the discussion of what studies were included in the literature review I would have appreciated more detail on why things were included or excluded at each stage. For instance what in the titles made you retain or cut a study? Etc.

Reviewer #2: Introduction

• Line 18: I suggest removing the phrasing that Pepin “coined” the terms, as this was not the first mention. It is fine to cite Pepin et al., but not as the primary origin.

Methods

• I appreciate the decision to simplify the model by removing some layers of complexity. This makes the analysis easier to interpret, but it would be worth noting in the discussion that RVF-endemic landscapes are highly heterogeneous, and simplification may limit applicability.

• The description of the Sobol methodology appears sound. While I am not a technical expert on Sobol indices, the explanation and approach are appropriate for the aims of the study.

• Please clarify your consideration of oviposition behavior: eggs are rarely laid in the middle of large water bodies. How was this handled in the model assumptions?

Reviewer #3: This work contains impressive modeling and sensitivity analysis of the models.

I am not skilled in evaluating this kind of model nor this kind of sensitive analysis.

However I do have substantial background in mosquito ecology and the epidemiology of Rift Valley fever virus.

There are assumptions in two of the authors’ key parameters (oviposition density and area scanned for oviposition sites) that are in my opinion not correct.

For oviposition density for both species it is not appropriate and potentially highly misleading to only consider surface area of the water body.

This is because the perimeter (i.e., total length) of the water’s edge is the biologically relevant factor.

A perfectly circular pond and a crenellated (star shaped) edge pond could have the exact same water area but have vastly different perimeters.

Perimeters are where the eggs are laid and so perimeters determine how many eggs are there.

Also, as a population, Aedes mosquitoes produce and oviposit multiple batches of eggs and continue to oviposit as the water recedes, right at the water’s edge.

Thus there could be multiple concentric perimeters of eggs at any given water body.

Also most of the parameterization in this model comes from Aedes Stegomyia (aegypti and albopictus) - neither are species currently of nigh importance in RVFV epidemiology in endemic areas.

Regarding scanning for oviposition sites (based on dispersal distance) - most of those studies from which the authors draw parameters were measuring dispersal of host-seeking females (newly emerged), not gravid blood fed females looking for oviposition sites.

There is no reason to believe that host seeking females would not return to the same water4 body they came from after blood feeding nearby.

Perhaps a more important parameter is distance to livestock host and the degree of ephemerality of the water body where immature development took place.

Finally, the authors really need to be quite explicit on how this work is going to inform future RVFV epidemiology research and operational relevance.

I am generally frustrated by RVFV modeling because the assumptions and parameterization are frequently fatally flawed - so please forgive my pointed criticism of this study.

Again, the modeling and mathematical capabilities of the authors might be beyond reproach, but in my view the biological realism is not as good as it could be and is likely misleading the modeling effort which could have real consequences if it is taken up into public health policy.

RVFV epidemiology in endemic areas is diverse:

The 2-species (Aedes/Culex) model only accounts for the major decadal epizootics and is really an East African phenomenon and seems to be less and less relevant compared to the burgeoning inter-epizootic nature of the virus throughout the potential transmission area in endemic lands.

Epidemiology of the virus in Southern Africa, Madagascar, Egypt, Arabian Peninsula, Western and Central Sahel — all very different, and especially to the dambo system in East Africa.

I urge the authors to target their modeling to advocate for mass larviciding in the mambo system and gather parameters and intervention modeling specific to this end.

Reviewer #4: The description of model structure and parameters is not sufficiently detailed. Readers unfamiliar with the earlier model might struggle to follow how mosquito dynamics are simulated. Parameter ranges and distributions used in the sensitivity analysis are not always clearly justified (why those ranges? based on literature, expert opinion, or assumptions?). The paper does not fully explain how Sobol indices were computed, which may limit reproducibility for readers outside the field. The Sobol indices are presented as aggregated across time, but mosquito ecology is inherently time-varying. This means the results miss important dynamics (e.g., rainfall may dominate early season, oviposition later). A time-dependent sensitivity analysis (Sobol indices as a function of time) would add much more ecological realism. The methods do not specify whether empirical calibration/validation of parameters was done using field data (e.g., mosquito counts, oviposition studies). Without at least partial validation, the model risks being too theoretical.

**Results**

-Does the analysis presented match the analysis plan?

-Are the results clearly and completely presented?

-Are the figures (Tables, Images) of sufficient quality for clarity?

Reviewer #1: The results are easy to follow, and the figures are nicely done. Currently the results for the sensitivity analysis are separated by culex and aedes but the discussion merges the two, as does the rest of the paper. It would have made more sense to me to separate these by parameters again and discuss both mosquito species at the same time under each parameter.

Reviewer #2: Results

• The results are well presented, with clear figures that effectively support the findings.

Reviewer #3: See above - problem with parameters

Reviewer #4: The objectives state the use of sensitivity analysis on ecological factors but do not clearly define all the parameters under investigation from the start (focus seems to emerge later on oviposition). The framing is broad (“factors influencing mosquito ecology”) but narrows abruptly to four parameters, which could confuse readers. While RVFV transmission is the context, the objectives do not clearly connect the ecological modeling to potential applications in vector control or outbreak preparedness. The results highlight that oviposition proportion is the most influential factor, but quantitative detail is limited. Sobol indices can capture interaction effects, but the manuscript mainly emphasizes main effects, leaving out valuable insights about parameter synergies. The constant vs. periodic environment scenarios are mentioned, but their comparative outputs are not deeply analyzed (e.g., how periodic changes alter sensitivity ranks over time).

**Conclusions**

-Are the conclusions supported by the data presented?

-Are the limitations of analysis clearly described?

-Do the authors discuss how these data can be helpful to advance our understanding of the topic under study?

-Is public health relevance addressed?

Reviewer #1: The conclusions seem fair. I found it interesting the the important parameters differed for the two mosquito species but discussion of this was scant. If the authors wish they could expand this part of the discussion for biologists.

Reviewer #2: Discussion

• Line 413: Please clarify the statement on disease spread via vector dispersal, noting that infected livestock movement was not considered in the model.

• If a major goal of the study is to guide future data collection; please be more explicit about which field data are most needed to refine the model and which factors should be prioritized in future sampling. Your paper is likely to have more (or just as many) field epidemiologist readers than modellers

• Line 430: Given the absence of oviposition data across water bodies, is it reasonable to assume the entire surface area is available for egg laying? Some comment on this assumption would be helpful.

• Line 482: Since mosquitos must feed on a host in order to produce eggs, is it acceptable to ignore this in the model? This should be acknowledged as a limitation.

• Please add a few lines discussing the overlap between ecological factors influencing oviposition and host availability, and whether this missing linkage constrains the model’s realism.

• I suggest adding a comment that this model could be useful for vector control programs targeting reductions in multiple mosquito species, not just Aedes spp. All need to lay eggs and feed

Reviewer #3: See above - problem with parameters

Reviewer #4: While the dominance of oviposition is valid, the discussion underplays the role of other parameters (temperature, water body fluctuations) and how they interact in real-world ecological systems. The paper calls for more field data but does not sufficiently engage with existing entomological studies on oviposition ecology in RVFV vectors. The discussion does not strongly connect the findings to disease surveillance or control strategies (e.g., how monitoring water bodies could inform early-warning systems). The contribution of the study (e.g., being one of the first to apply global sensitivity analysis to RVFV mosquito ecology) could be highlighted more clearly to strengthen the impact.

**Editorial and Data Presentation Modifications?**

Reviewer #1: I would present the 4 parameters you compared, in two species (so really 8 parameters) in the abstract/intro and not just vaguely say 4 parameters. I had to read all the way through the results to find what parameters for mosquitos you were going to use in the sensitivity analysis. Also it was not clear to me until the results that you did 4 parameters in two species of mosquitoes - and the results differed by species so consider clarifying this in the abstract.

Reviewer #2: (No Response)

Reviewer #3: Comma after i.e. and e.g.

Check italicization of species names.

Reviewer #4: The manuscript entitled “Sensitivity analysis of factors influencing the ecology of mosquitoes involved in the transmission of Rift Valley Fever Virus” presents a deterministic compartmental model to evaluate the sensitivity of ecological parameters that shape mosquito ecology. By applying Sobol global sensitivity analysis and literature-based parameter ranges, the authors identify influential parameters and highlight essential knowledge gaps, particularly regarding oviposition site use. This approach is methodologically sound and relevant, as it advances understanding of the ecological drivers underlying Rift Valley fever transmission dynamics. However, while the Sobol global sensitivity analysis provides useful insights into the relative importance of ecological parameters, the current approach treats sensitivity as static across the entire simulation horizon. Given that mosquito ecology and RVFV transmission risk are highly seasonal and time-dependent, I strongly recommend that the authors consider conducting a time-dependent (dynamic) sensitivity analysis. This would allow them to capture temporal shifts in parameter influence (e.g., rainfall dominating early in the season vs. oviposition later), which is crucial for both ecological understanding and practical vector control planning.

If full implementation is not feasible within the current submission, at a minimum the authors should discuss the limitations of a static approach and outline how time-dependent sensitivity analysis could improve the robustness of future modeling work.

Here are minor issues that the authors should address to clarify specific methodological points:

1) The abstract does not clearly articulate the public health implications of the findings. As written, it gives the impression that the contribution is primarily methodological.

2) The literature review focuses exclusively on Aedes aegypti. The authors must justify this choice, whether it is due to the relevance of this species in the Kenyan context or a lack of species-specific data for other vectors.

3) The rationale for selecting the four specific parameters over other potential model parameters needs further elaboration. The authors explain that these four are the most uncertain, but is that alone sufficient to justify excluding other parameters that may also influence the ecology?

4) Justify the choice of parameter distribution (uniform distribution) for \kappa and \rho. Is this biologically defensible?

5) Discuss the convergence of the Sobol’s indices.

6) Quantifying and discussing this interaction (e.g., does the importance of egg density depend on the proportion of area available?) would provide much richer ecological insight. This can be undertaken by exploring other orders.

7) The constant vs. periodic environment scenarios are mentioned, but their comparative outputs are not deeply analyzed (e.g., how periodic changes alter sensitivity ranks over time).

8) The discussion does not leverage key results with essential implications (e.g. the differential sensitivity of Culex and Aedes to water body size should be discussed in the context of habitat targeting for control)

9) The outcomes from the sensitivity analysis imply that factors affecting vector fecundity are less critical than habitat structure?

10) Host detection is not a limiting factor in the current model. The authors should discuss whether this holds under the condition of low livestock density.

**Summary and General Comments**

Reviewer #1: This manuscript is interesting and useful to folks modelling vector-borne disease - specifically RVF. I would have appreciated some discussion about whether the authors think the parameters that are important in the sensitivity analysis would hold in other similar vector-borne diseases or if they think this is RVF specific?

Reviewer #2: This manuscript presents a sensitivity analysis of ecological factors influencing mosquito dynamics relevant to RVFV transmission. Using a deterministic, compartmental model developed for RVFV in Kenya, the authors examine the relative influence of parameters such as oviposition area, egg density, and dispersal on mosquito abundance.

The study is well presented and addresses an important gap in understanding how environmental conditions shape vector populations. The figures are clear and the methodological approach, while simplified, is accessible and informative. The main strength is the identification of oviposition area as a highly influential parameter, highlighting the need for targeted field data. I would suggest the authors to think about what this might mean for future vector control programmes and how they think this model fits within the broader field since it has been submitted to a multidisciplinary journals and readers will ask “what does this mean, practically?”

My review focuses on clarifications, limitations, and suggestions to strengthen the manuscript’s impact. I am not an expert on the modelling methodology but the methods appear sound.

Key overall points:

• Please use spp. when referring to mosquito genera.

• Emphasize in the limitations that vertical transmission and hatching of Aedes eggs is not the only way RVFV is maintained between outbreaks; livestock movements and secondary vectors (e.g., Culex, Mansonia) are also important. Why make this distinction if you combine them for the model? Your main point seems to be that both are important for transmission, but they have different life history traits -- which I would agree with

• Consider discussing that breeding is not limited to water bodies alone; other oviposition sites matter, and water body size may also act as a proxy for rainfall variability.

• This is a very useful summary of mosquito parameters relevant for spatial analysis.

• Ground some of the ecological assumptions more explicitly (e.g., does oviposition occur the middle of water bodies or mostly around the edge? ).

• I like that you say data collection needs to improve for parameterization, but strengthen the impact of the study by specifying exactly what new data are most needed to improve models like yours and which ecological factors should be prioritized in sampling approaches.

• While not required for publication, I do notice that no coauthors are from institutes in Kenya or East Africa. Such collaborations would be very useful to address some of the gaps around impact of the manuscript and implications for practical control programmes.

Reviewer #3: Problem with parameters

Reviewer #4: The manuscript entitled “Sensitivity analysis of factors influencing the ecology of mosquitoes involved in the transmission of Rift Valley Fever Virus” presents a deterministic compartmental model to evaluate the sensitivity of ecological parameters that shape mosquito ecology. By applying Sobol global sensitivity analysis and literature-based parameter ranges, the authors identify influential parameters and highlight essential knowledge gaps, particularly regarding oviposition site use. This approach is methodologically sound and relevant, as it advances understanding of the ecological drivers underlying Rift Valley fever transmission dynamics. However, while the Sobol global sensitivity analysis provides useful insights into the relative importance of ecological parameters, the current approach treats sensitivity as static across the entire simulation horizon. Given that mosquito ecology and RVFV transmission risk are highly seasonal and time-dependent, I strongly recommend that the authors consider conducting a time-dependent (dynamic) sensitivity analysis. This would allow them to capture temporal shifts in parameter influence (e.g., rainfall dominating early in the season vs. oviposition later), which is crucial for both ecological understanding and practical vector control planning.

If full implementation is not feasible within the current submission, at a minimum the authors should discuss the limitations of a static approach and outline how time-dependent sensitivity analysis could improve the robustness of future modeling work.

PLOS authors have the option to publish the peer review history of their article (what does this mean?). If published, this will include your full peer review and any attached files.). If published, this will include your full peer review and any attached files.). If published, this will include your full peer review and any attached files.). If published, this will include your full peer review and any attached files.

...

Reviewer #1: No

Reviewer #2: No

Reviewer #3: No

Reviewer #4: No

**Figure resubmission:**
---

## [Decision Letter · Decision Letter 1]

26 Feb 2026

Response to Reviewers'. This file does not need to include responses to any formatting updates and technical items listed in the 'Journal Requirements' section below.'. This file does not need to include responses to any formatting updates and technical items listed in the 'Journal Requirements' section below.* A marked-up copy of your manuscript that highlights changes made to the original version. You should upload this as a separate file labeled 'Revised Manuscript with Track Changes'.'.* An unmarked version of your revised paper without tracked changes. You should upload this as a separate file labeled 'Manuscript'.'.If you would like to make changes to your financial disclosure, competing interests statement, or data availability statement, please make these updates within the submission form at the time of resubmission. Guidelines for resubmitting your figure files are available below the reviewer comments at the end of this letter.We look forward to receiving your revised manuscript.Kind regards,Ran Wang, M.D.Academic EditorPLOS Neglected Tropical DiseasesAudrey LenhartSection EditorPLOS Neglected Tropical Diseases

Shaden Kamhawi

co-Editor-in-Chief

Paul Brindley

co-Editor-in-Chief

**Reviewers' comments:**Reviewer's Responses to Questions

**Key Review Criteria Required for Acceptance?**

**Methods:**

-Are the objectives of the study clearly articulated with a clear testable hypothesis stated?

-Is the study design appropriate to address the stated objectives?

-Is the population clearly described and appropriate for the hypothesis being tested?

-Is the sample size sufficient to ensure adequate power to address the hypothesis being tested?

-Were correct statistical analysis used to support conclusions?

-Are there concerns about ethical or regulatory requirements being met?

Reviewer #5: Yes.

Reviewer #6: The methods used for the literature review were adequate. The model and parameters and sensitivity analysis methods were well explained for readers of different backgrounds.

Reviewer #7: Line 169: How do you define the typical area scanned by a mosquito before finding an oviposition site? I see that you define it later, but please elaborate further here to provide context.

Line 276: Missing word (“of”?)

Reviewer #8: (No Response)

**Results:**

-Does the analysis presented match the analysis plan?

-Are the results clearly and completely presented?

-Are the figures (Tables, Images) of sufficient quality for clarity?

Reviewer #5: Yes.

Reviewer #6: The results were linked to the aim and objective of the analysis. Referral was made to the mathematical model that was previously developed. The tables and figures were very informative and formatted according to best practices in writing standards.

Reviewer #7: Line 314: Add the caveat that Aedes albopictus are container/treehole mosquito species, not a flood pan/dambo species, so dispersal behaviour may be different. I would also reorganize this paragraph to list the Aedes spp. together and the Culex spp. together.

Line 371: Was egg production of wild and colonised-strains compared in this study? Or are you comparing to the study in line 369 of blood-deprived mosquitoes (which arguably is not a sound comparison of lab vs. field)? Please clarify.

Line 384: Again you should explicitly state the different ecologies of container breeders, like Ae. aegpyti and albopictus, and floodwater species like mcintoshi.

Line 381: Perhaps not particularly surprising - bloodmeals are more closely linked to fertility than sugar-feeding in mosquitoes.

Line 396: What do you mean by nutritional status? Do you mean larval diet? Adult body mass? Bloodmeal volume/frequency? Please clarify.

Reviewer #8: (No Response)

**Conclusions:**

-Are the conclusions supported by the data presented?

-Are the limitations of analysis clearly described?

-Do the authors discuss how these data can be helpful to advance our understanding of the topic under study?

-Is public health relevance addressed?

Reviewer #5: Yes.

Reviewer #6: Following the previous review, the authors made good conclusions with recommendations that specified gaps in data for model parameters.

Reviewer #7: Line 577: Egg production may not be the best indicator of population size, because only a small proportion of these actually survive to adulthood. Please elaborate on the decision to focus on the egg stage. Also, you have not considered competition in this model – some mosquitoes avoid habitats that already have a high density of conspecific eggs or larvae.

Line 739: Although collecting more data is certainly a way of addressing uncertainty, it’s worth acknowledging the challenges of directly collecting data for certain mosquito ecological/biological parameters (e.g. lifespan).

Reviewer #8: (No Response)

**Editorial and Data Presentation Modifications?**

Reviewer #5: None.

Reviewer #6: (No Response)

Reviewer #7: Lines 8-9: I would argue that this isn’t a modelling limitation so much as a data limitation (which the authors themselves point out in this paper). Consider rephrasing or adding that caveat to this sentence.

Line 11: Is Garissa a state? Province? Indicate so that the scale of the outbreak is in context.

Line 21: Please specify that these are floodwater Aedes spp.

Line 23: Their vector competence and role in transmission remains unclear, so I would change this to “may be involved”.

Line 54: I don’t quite understand what you mean by “are not subject to arbitrary variation”. Pleased clarify this sentence.

Please improve the resolution of all figures, particularly Figures 4 and 5, which are quite blurry.

Reviewer #8: (No Response)

**Summary and General Comments:**

Reviewer #5: This manuscript presents a detailed deterministic compartmental model to evaluate the sensitivity of ecological parameters that shape the ecology of Culex and Aedes mosquitoes in the context of Rift Valley Fever Virus (RVFV) transmission. The authors clearly present their introduction, methods, results and discuss their findings, highlight current research gaps and list the limitations of the study. Furthermore, they have appropriately responded to the previous review comments. Although this manuscript is highly mathematical as it is model-based, they clearly present its findings in a way that can be used by people implementing RVFV prevention/control strategies and by researchers to fill up the existing gaps.

Reviewer #6: The authors conducted a sensitivity analysis of ecological input parameters that influence mosquito dynamics and, consequently, Rift Valley fever transmission patterns using a deterministic, compartmental model specifically designed for RVFV in Kenya. To determine the parameter values, they reviewed existing literature. This is a very useful study. The model focuses on ecology and mosquitoes, thereby contributing to understanding and predicting RVF outbreaks. However, because the RVF eco-epidemiology is very complex, involving a broad host system of livestock and wildlife, it has limitations. Following the review of the previous manuscript, the authors clearly defined and elaborated on these limitations. The model is also hindered by a lack of field data on certain input parameters, and they explain what further data collection is needed to better understand the value and distribution of these variables. The methods and results are well presented, and the text is generally clear, easy to follow, and flows smoothly. I think the authors did a very good job tackling this challenging analysis. This publication could significantly contribute to improving the prediction of RVF outbreaks when combined with findings from other studies focusing on factors such as horizontal transmission patterns and immunity variability in ruminant livestock and wildlife and historical data on animal and human cases.

Reviewer #7: The authors present a variance-based sensitivity analysis of the ecological drivers of RVFV vector population dynamics. Although I am not overly familiar with Sobol analysis, I believe that the work provides valuable insights into the potential processes underpinning RVFV transmission. I have only a few comments that I would like to see addressed before the manuscript is published.

Reviewer #8: (No Response)

PLOS authors have the option to publish the peer review history of their article (what does this mean?). If published, this will include your full peer review and any attached files.). If published, this will include your full peer review and any attached files.). If published, this will include your full peer review and any attached files.). If published, this will include your full peer review and any attached files.

...

Reviewer #5: No

Reviewer #6: No

Reviewer #7: No

Reviewer #8: No

**Figure resubmission:**While revising your submission, we strongly recommend that you use PLOS’s NAAS tool (https://ngplosjournals.pagemajik.ai/artanalysis) to test your figure files. NAAS can convert your figure files to the TIFF file type and meet basic requirements (such as print size, resolution), or provide you with a report on issues that do not meet our requirements and that NAAS cannot fix.

**Reproducibility:**To enhance the reproducibility of your results, we recommend that authors of applicable studies deposit laboratory protocols in protocols.io, where a protocol can be assigned its own identifier (DOI) such that it can be cited independently in the future. Additionally, PLOS ONE offers an option to publish peer-reviewed clinical study protocols. Read more information on sharing protocols at https://plos.org/protocols?utm_medium=editorial-email&utm_source=authorletters&utm_campaign=protocolsTo enhance the reproducibility of your results, we recommend that authors of applicable studies deposit laboratory protocols in protocols.io, where a protocol can be assigned its own identifier (DOI) such that it can be cited independently in the future. Additionally, PLOS ONE offers an option to publish peer-reviewed clinical study protocols. Read more information on sharing protocols at https://plos.org/protocols?utm_medium=editorial-email&utm_source=authorletters&utm_campaign=protocols

---

## [Editor Report · Decision Letter 2]

25 Mar 2026

Dear Dr Boëte,

We are pleased to inform you that your manuscript 'Sensitivity analysis of factors influencing the ecology of mosquitoes involved in the transmission of Rift Valley fever virus' has been provisionally accepted for publication in PLOS Neglected Tropical Diseases.

Best regards,

Ran Wang, M.D.

Academic Editor

Audrey Lenhart

Section Editor

Shaden Kamhawi

co-Editor-in-Chief

Paul Brindley

co-Editor-in-Chief

---

## [Editor Report · Acceptance letter]

Dear Dr Boëte,

We are delighted to inform you that your manuscript, "Sensitivity analysis of factors influencing the ecology of mosquitoes involved in the transmission of Rift Valley fever virus," has been formally accepted for publication in PLOS Neglected Tropical Diseases.

Best regards,

Shaden Kamhawi

co-Editor-in-Chief

Paul Brindley

co-Editor-in-Chief
